# Physics-informed deep learning characterizes morphodynamics of Asian soybean rust disease

Henry Cavanagh[1], Andreas Mosbach[2], Gabriel Scalliet [2], Rob Lind[3] & Robert G. Endres [1✉]

Medicines and agricultural biocides are often discovered using large phenotypic screens across hundreds of compounds, where visible effects of whole organisms are compared to gauge efficacy and possible modes of action. However, such analysis is often limited to human-defined and static features. Here, we introduce a novel framework that can characterize shape changes (morphodynamics) for cell-drug interactions directly from images, and use it to interpret perturbed development of *Phakopsora pachyrhizi*, the Asian soybean rust crop pathogen. We describe population development over a 2D space of shapes (morphospace) using two models with condition-dependent parameters: a top-down Fokker-Planck model of diffusive development over Waddington-type landscapes, and a bottom-up model of tip growth. We discover a variety of landscapes, describing phenotype transitions during growth, and identify possible perturbations in the tip growth machinery that cause this variation. This demonstrates a widely-applicable integration of unsupervised learning and biophysical modeling.

[1] Centre for Integrative Systems Biology and Bioinformatics, Imperial College London, London SW7 2BU, UK. [2] Syngenta Crop Protection AG, Schaffhauserstrasse 101, 4332 Stein, Switzerland. [3] Syngenta International Research Centre, Jealott's Hill, Berkshire RG42 6EY, UK.
✉email: r.endres@imperial.ac.uk

Quantifications of cell shape changes (morphodynamics) can reveal key developmental transitions and behavioral strategies, as well as modes of action of drugs by comparison with known drug-phenotype mappings[1,2]. Although recent progress in automated image analysis has popularized static descriptors beyond mean growth rates and metabolic fluxes[3], the incorporation of dynamics can provide more complete system descriptions[4] and may also aid the development and validation of mechanistic models[5]. Here, we developed such a framework and used it to interpret how fungicides affect the morphodynamics of *Phakopsora pachyrhizi*, the pathogen that causes rust disease in the soybean crop worldwide. Spores land on soybean leaves, grow germ tubes, penetrate the plant using appressoria, and subsequently form haustoria that extract nutrients[6], as sketched in Fig. 1a. This little-understood pathogen can cause economically devastating yield losses of up to 80%[7] and is fast resistance evolving[8].

Current methods for quantifying organism morphodynamics typically rely on a low-dimensional space of interpretable features[9]. Although existing methods have uncovered remarkable behavioral patterns, revealing chemotactic strategies, temporal processing, and social cooperation in a range of organisms[10–13], typical shortcomings are as follows: first, they are often based on particular shape descriptors (e.g., one-dimensional centerlines), restricting analysis to a narrow range of morphologies, often requiring sophisticated feature extraction algorithms. Second, they focus on stereotyped behaviors, which may not be characteristic of early development. Finally, states are often discretized and transition probabilities extracted, which obscures the continuous nature of morphodynamics. Alternatively, statistical analysis can be based on thousands of behavioral features to cluster and classify compounds, and compare the results with known modes of action, but results can be difficult to interpret[2]. In contrast to statistical correlates, interpretable continuous and stochastic models can be considered, but are not yet associated with morphodynamics. For instance, in Waddington's epigenetic landscape, cells begin at the top as pluripotent and subsequently develop into a number of differentiated states, represented as lower-level valleys[14]. Despite many applications[15–19], to our knowledge, such landscapes have only been uncovered at steady state.

A major method for image analysis, dimensionality reduction, and inference is deep learning[20]. Feedforward deep neural networks are composed of a series of interconnected layers of artificial neurons, which transform inputs through a series of nonlinear operations[21]. Network parameters are updated through stochastic gradient descent and its variants in order to minimize a human-defined objective (the loss). These networks have been proven capable of approximating any function, given a sufficient number of parameters[22]. To link data analysis and modeling, two types of network are particularly important: first, autoencoders, which are capable of capturing low-dimensional structure in data through a constrained reconstruction task[23]. Due to their high flexibility, autoencoders have many advantages over non-parametric *t*-distributed stochastic neighbor embedding and linear principal component analysis algorithms. Second, physics-informed neural networks (PINNs). Although forward problems can be solved fast with grid-based methods, PINNs can solve inverse (inference) problems for a wide range of differential equations[24,25].

Here, we aimed to extract morphodynamic characterizations for intuitive comparison across compounds. Specifically, we analyzed *P. pachyrhizi*, germinating in vitro in a control compound and several fungicides, by first using an autoencoder to uncover a single two-dimensional (2D) morphospace of salient features from high-throughput images of fixated fungi (Fig. 1a, b). We then fitted two minimal models of dynamics over this morphospace and took their condition-dependent parameters as informative characterizations (Fig. 1c, d). The two models approach the dynamics from opposite directions: the first is a top-down model that utilizes a Fokker–Planck description to uncover the global morphodynamic driving forces in the form of Waddington-type landscapes, and the second is a bottom-up persistent random walk model of the growth zone at the tip. We used a PINN to infer the landscapes and approximate Bayesian computation (ABC), in conjunction with a morphospace-derived similarity metric, to infer the parameter posteriors of the tip growth model. The uncovered landscapes show that morphodynamics are diffusion-dominated until the germ tube begins to bend, at which point deterministic forces begin to drive trajectories apart. Fungicide-induced deformations include barriers, plateaus, and canalized pathways, which may arise from differing stabilities of the growth zone. To avoid "black-box" methods, we analyzed our PINN to interpret the convergence. Our work shows how intuitive system characterizations can be acquired directly from images, by integrating unsupervised learning and biophysical modeling.

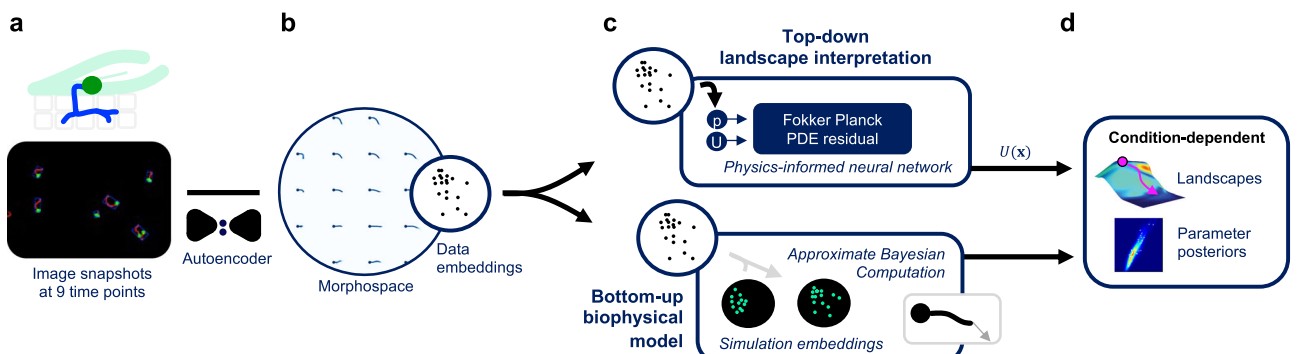

**Fig. 1 Morphodynamics of the Asian soybean rust pathogen, *P. pachyrhizi*, are characterized through condition-dependent dynamics over a global morphospace. a** *P. pachyrhizi* burrows into soybean leaves to extract nutrients, as sketched (top). Image sets at nine time points under six conditions (bottom) are processed to yield aligned, single-fungus images. **b** An autoencoder learns the biophysical degrees of freedom from the images, discovering a 2D morphospace. **c** Dynamics are characterized using two models: a top-down landscape ($U(\mathbf{x})$) model, where a physics-informed neural network fits the Fokker–Planck equation to the morphospace embeddings, and a bottom-up persistent random walk model of the growth zone, with parameters fitted using approximate Bayesian computation with a morphospace-derived similarity metric. These yield **d** interpretable, condition-dependent characterizations in the form of Waddington-type landscapes and tip growth parameter posteriors.

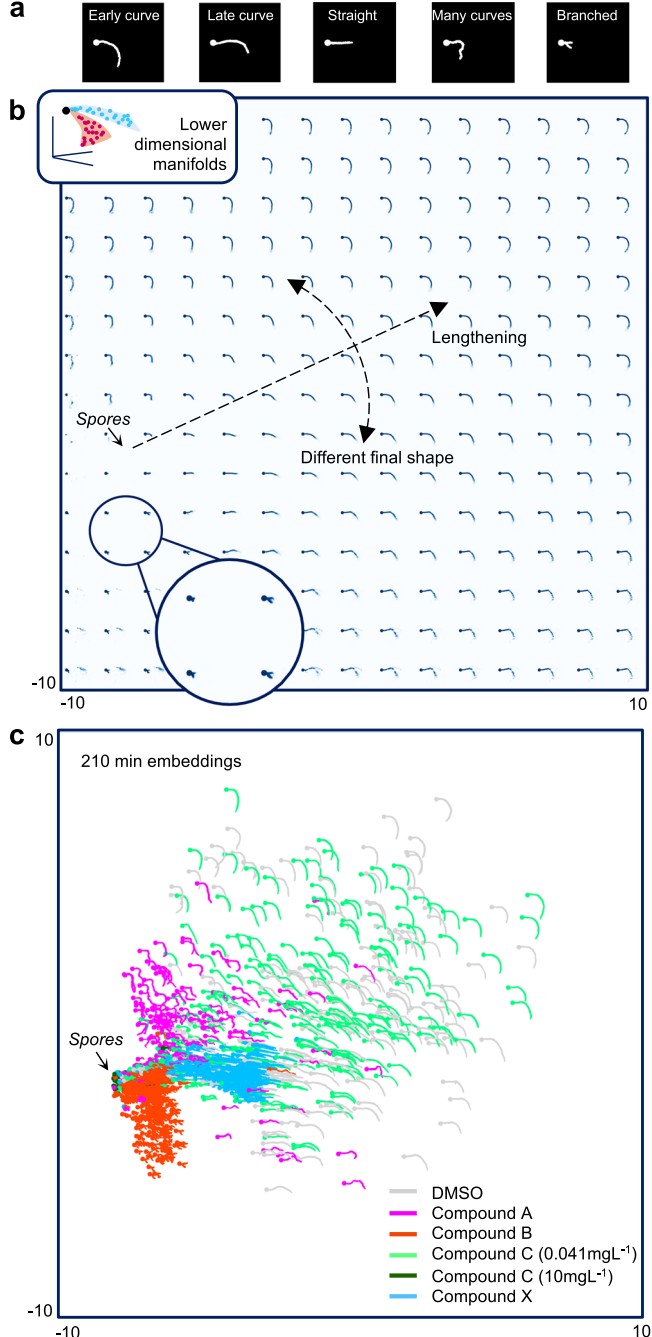

**Fig. 2 Global morphospace learned by a convolutional autoencoder.**
**a** Human categorization of *P. pachyrhizi* phenotypes is time consuming and introduces human biases and unnatural discretization. **b** A convolutional autoencoder addresses these shortcomings by learning the manifold associated with each condition, sketched inset. Morphospace features are shown by propagating morphospace coordinates on a grid through the decoder. **c** 210 min embeddings for all conditions show that fungicides can induce perturbed dynamics over the morphospace, which therefore represents for an expressive space for differentiating morphodynamics upon.

## Results

**A global morphospace reveals perturbed morphodynamics.**
Manual categorization of phenotypes is time consuming and limited to discrete human-defined features (Fig. 2a). In contrast, manifold-based dimensionality reduction can provide a continuous low-dimensional space where dynamics are as simple as

possible[26]. This is because an imaged dynamic system with *n* degrees of freedom traces out an *n*-dimensional manifold within the higher-dimensional pixel space, irrespective of the image dimensionality.

To learn such a morphospace for *P. pachyrhizi* growth, we carried out a high-throughput imaging assay of distinct populations at nine equally spaced time points, between 90 and 210 min after mixing with different compounds (code names used henceforth given in brackets): a dimethylsulfoxide control (DMSO), methyl benzimidazol-2-ylcarbamate at 1.1 mg L$^{-1}$ (carbendazim, Compound A), PIK-75 hydrochloride at 3.3 mg L$^{-1}$ (Compound B), benzovindiflupyr (Compound C) at 0.041 and 10 mg L$^{-1}$, and Compound X (a Syngenta research compound related to trifluoromethyloxadiazoles[27]; see Supplementary Fig. 7 for the chemical structure) at 1.1 mg L$^{-1}$. These compounds were identified at pre-screening to show a range of phenotypes. We henceforth refer to each combination of compound and concentration as a condition. We extracted single-fungus images from the snapshot data sets, aligned such that the initial growth directions coincided, using automated processing, which yielded ~600,000 images in total (with mean and SD across snapshots of ~11,000 and 3000, respectively). In order to validate the inferred Fokker–Planck model parameters and to motivate the tip growth model, we also gathered a small number of time-lapse videos of individual fungi for each compound (3–8, see Supplementary Movies 1–5) and aligned these manually. High-throughput analysis is not carried out with time-lapse videos in industry due to technical limitations. See "Methods" and Supplementary Note 1 for details on the compounds, imaging and image processing, and Supplementary Fig. 4a for an example time-lapse sequence.

The morphodynamics are perturbed differently in different conditions and so the manifold traced out by a *P. pachyrhizi* population is condition-dependent (as sketched in the inset of Fig. 2b). To pull the condition-dependent manifolds together into a global morphospace, we trained a convolutional autoencoder (CAE)[28], an architecture specialized for images, with a 2D code space on images from all conditions and times (see Supplementary Note 1 for details on the architecture and training). To ensure the CAE used the morphodynamic manifolds to solve this reconstruction task, we selected the network complexity and training time that provided the simplest trajectories for embeddings associated with a small number of single-fungus videos.

Figure 2b shows the distribution of features over the morphospace, found by propagating morphospace coordinates on a grid through the decoder. Distance from the spore embedding loosely captures lengthening and angle captures variation in final shape. Figure 2c shows the 210 min embeddings for all conditions, revealing how fungicides can perturb dynamics over the morphospace. Some introduce novel features, e.g., the branching by Compound B, whereas others change the distribution over wild-type features, e.g., the increased prevalence of straightening induced by Compound X. To move from coarse-grained qualitative insights to quantitative characterizations, we next fitted two simple models of dynamics over the morphospace.

**Emergent landscapes show the morphodynamic driving forces.**
Condition-dependent Waddington-type landscapes are intuitive morphodynamic characterizations that can be inferred from the snapshot embeddings evolving over the morphospace. We model each condition as inducing a time-independent field of driving forces, $\mathbf{F}(\mathbf{x}; c, \boldsymbol{\lambda})$, which depends on the compound concentration, $c$, and any pharmacophores, captured in a parameter vector, $\boldsymbol{\lambda}$. In the absence of any curl, as we assume for early development, the force field can be associated with the gradient of a quasi-potential, $U$, via $\mathbf{F} = -\nabla U$, which we take as the developmental landscape.

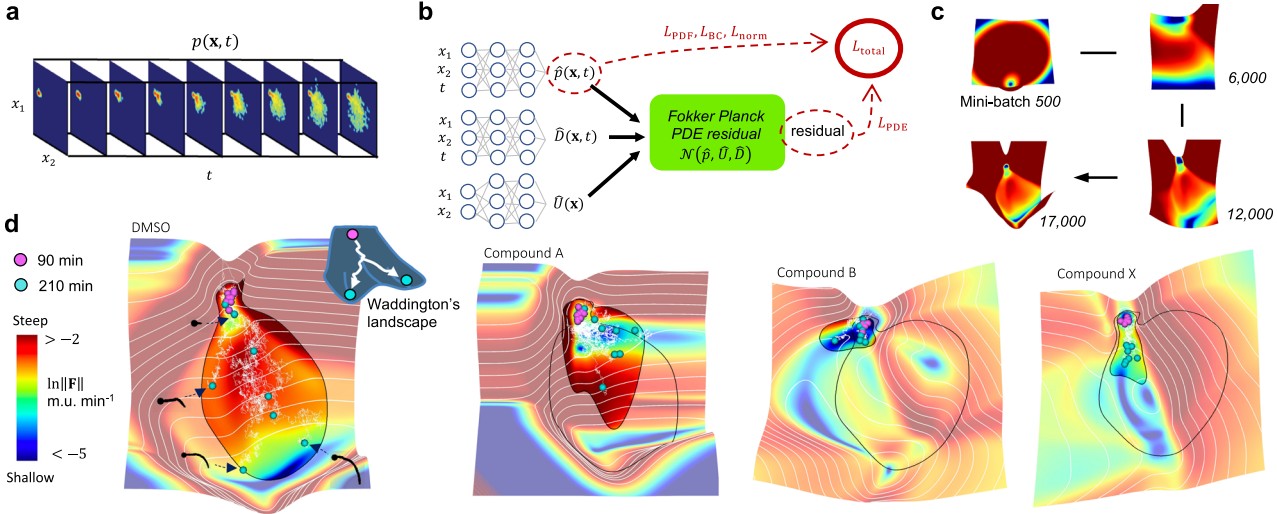

**Fig. 3 Morphodynamic landscapes learned by the PINN. a** Morphospace embeddings are transformed into probability density functions (PDFs), $p(\mathbf{x}, t)$, using kernel density estimation (KDE), yielding nine snapshots per condition. **b** A physics-informed neural network (PINN) learns the landscapes by fitting the Fokker–Planck equation to the PDFs. For each condition, the architecture comprises a neural network to learn each of the PDF, $\hat{p}(\mathbf{x}, t)$, diffusivity, $\hat{D}(\mathbf{x}, t)$, and landscape, $\hat{U}(\mathbf{x})$. The outputs of these are put through a series of differential operators that outputs the Fokker–Planck residual, $\mathcal{N}$, and the architecture is trained to match the data ($L_{\text{PDF}}$), minimize the magnitude of the residual ($L_{\text{PDE}}$), satisfy the boundary conditions ($L_{\text{BC}}$), and learn a normalized PDF ($L_{\text{norm}}$). **c** The architecture is trained over a series of mini-batches, with lower-frequency solutions explored first. **d** Landscapes with simulated particles, from 90 min (pink) to 210 min (blue) after mixing with compounds, are shown, colored by the gradient magnitude, $\|\mathbf{F}\|$ (in terms of morphospace units, m.u.). These are analogous to Waddington's epigenetic landscape, as sketched in the inset. The black outlines show the contour where the PDF learned by the PINN is $10^{-3}$ for DMSO and each condition. The inner region therefore highlights areas with high data density, with the remaining areas shown to facilitate connection with the morphospace and outer tendencies. Contour lines are plotted along equal landscape values, with spacings of 0.11, 0.08, 0.07, and 0.09 m.u.$^2$ min$^{-1}$ for the landscapes from left to right. Morphodynamics are diffusion-dominated until the germ tube begins to bend, at which point deterministic forces begin to drive trajectories apart. Fungicide-induced deformations including barriers, plateaus, and canalized pathways. This susceptibility to deformation, combined with the generality of the model, make the Fokker–Planck model well-suited for system characterization.

For cases where the underlying force field does have a curl, landscapes can still be uncovered by splitting the force into curl-free and curl-containing components, yielding a "potential and flux" description[15].

To enable the inference of landscapes, we connected the evolving snapshot embeddings to the driving forces through a Fokker–Planck model. These embeddings were transformed into probability density functions (PDFs) on a grid using kernel density estimation (KDE, Fig. 3a)[29]. The Fokker–Planck partial differential equation (PDE) is used to separate out a system's driving forces and stochastic processes, and model statistical ensembles of Brownian particles. Each particle moves over the landscape according to the following stochastic differential equation:

$$\mathbf{dx} = -\nabla U(\mathbf{x})dt + \boldsymbol{\sigma}(\mathbf{x}, t)\mathbf{dW}, \tag{1}$$

where $\mathbf{x}$ and $t$ are the position in 2D space and time, $\boldsymbol{\sigma}(\mathbf{x}, t)$ is a noise matrix, and $\mathbf{dW}$ is the Wiener process. $\boldsymbol{\sigma}(\mathbf{x}, t)$ has diagonal entries $\sqrt{2D(\mathbf{x}, t)}$ and zeros elsewhere, with $D$ being the diffusivity. The Fokker–Planck equation for the evolution of the PDF of the particle positions, $p(\mathbf{x}, t)$, is then

$$\frac{\partial p(\mathbf{x}, t)}{\partial t} = \sum_{i=1}^{2} \frac{\partial}{\partial x_i} \left[ \frac{\partial U(\mathbf{x})}{\partial x_i} p(\mathbf{x}, t) + \frac{\partial}{\partial x_i} [D(\mathbf{x}, t) p(\mathbf{x}, t)] \right]. \tag{2}$$

To learn the Fokker–Planck landscapes (and diffusivities) given the PDF data, i.e., solve the inverse problem, we used a PINN[24]. These learn PDE solutions by optimally matching PDF data, minimizing the magnitude of the PDE residual, and satisfying any further constraints, e.g., boundary conditions. PINNs have several favorable properties over alternative methods for solving the inverse problem[25]. First, in going from the forward to the inverse

problem, the only change required is the addition of extra learnable parameters; second, they can infer the governing equation with only sparse data, as they solve the inference of the full PDF and governing equation as a joint task[30]; third, they learn a continuous fully differentiable solution, which means other variables of interest can be calculated directly from the learned variables, without numerical approximation (e.g., useful when transitioning between potential and force); and fourth, they learn progressively more complex functions as training progresses. As we show, this is a useful property when combined with early stopping if the required function complexity is not known a priori. Finally, they scale more favorably with system dimensionality than grid-based methods, which can often perform well only for low-dimensional problems. This final property will prove especially useful when extending this work to higher-dimensional morphospaces.

We used one network to learn each of the PDF, diffusivity, and potential (Fig. 3b). The outputs of these ($\hat{p}(\mathbf{x}, t)$, $\hat{D}(\mathbf{x}, t)$ and $\hat{U}(\mathbf{x})$, respectively) are put through a series of differential operators ($\mathcal{N}$) that outputs the Fokker–Planck residual,

$$\mathcal{N}(\hat{p}, \hat{U}, \hat{D}) = -\frac{\partial \hat{p}}{\partial t} + \sum_{i=1}^{2} \frac{\partial}{\partial x_i} \left[ \frac{\partial \hat{U}}{\partial x_i} \hat{p} + \frac{\partial}{\partial x_i} (\hat{D}\hat{p}) \right], \tag{3}$$

which is incorporated into the loss so that the solution obeys the PDE.

The total loss to be minimized ($L_{\text{total}}$) is the sum of four terms (shown in full in "Methods"). The first three are calculated over randomized mini-batches. They are the mean-squared error between the data and learned PDF ($L_{\text{PDF}}$), the mean-squared PDF at the boundary ($L_{\text{BC}}$), and the mean-squared PDE residual ($L_{\text{PDE}}$). The final term ensures the PDF is normalized and is the squared difference between unity and a numerical integration

over the full grid at a randomly selected time point ($L_\mathrm{norm}$). The relative importance of these terms is determined by hyperparameters $a$, $b$, $c$, and $d$,

$$L_\mathrm{total} = aL_\mathrm{PDF} + bL_\mathrm{BC} + cL_\mathrm{PDE} + dL_\mathrm{norm}, \qquad (4)$$

and lower-frequency functions are explored first (Fig. 3c[31]), in alignment with Occam's razor. An ablation analysis also confirms that both space and time-dependent diffusion are required for the best model fit (Supplementary Fig. 1a).

Letting the PINN train to convergence would result in overfitting. This is the phenomenon where a neural network's function pushes beyond the problem-dependent desired complexity; e.g., in image classification, the network begins to learn spurious patterns and to generalize poorly to new data. In the context of inference from independent snapshots, overfitting corresponds to fitting to differences between individual snapshots that arise from variability between *P. pachyrhizi* batches. Although having independent snapshots is beneficial in that it shows a wider range of dynamics, this can result in features such as a non-monotonically decreasing fraction of spores, which should not be captured in the model. The PINN learns trends common to all snapshots first and we stop training before overfitting begins, identified by monitoring the spore region (as shown in Supplementary Fig. 1b, c for three training repeats). This regularization technique is known as early stopping. We note that the loss exponentially decays after an initial period of fast improvement. Hence, reasonable results can be achieved using much earlier stopping points than used here, if computational time is limited. Further details on the PINN architecture, hyperparameters, and optimization can be found in "Methods" and Supplementary Note 2.

Figure 3d shows the landscapes learned by the PINN, as well as a sample of forward simulations of Eq. (1) (see "Methods" for details on the forward simulations), and the correspondence between the landscapes and morphospace is shown in Supplementary Fig. 2. The overfitting point described above approximately corresponds to 30, 30, 30, 20, 25, and 25 h of training for DMSO and Compounds A, B, C (0.041 mg L$^{-1}$), C (10 mg L$^{-1}$), and X, respectively. Diffusion over the landscapes has two sources: (i) morphodynamic diffusion, i.e., the fundamental unpredictability of morphology change from one time step to the next on the 2D morphological data manifold; and (ii) embedding noise, which arises because variations in image resolution, segmentation, and alignment induce perturbations away from the manifold, and the complexity of the autoencoder's embedding function means these perturbations are not always just mapped to the closest point on the 2D manifold. As the same image pre-processing algorithms were used on all conditions, differences in the Fokker–Planck diffusion over the same region of morphospace will be morphodynamic in nature rather than arising from the embedding noise.

The condition-dependent landscapes are highly interpretable (Fig. 3d). DMSO development begins in a metastable region of spores where diffusion dominates, with a threshold energy required for germination, and annealing diffusivity capturing a subpopulation of spores that never cross this threshold (Supplementary Fig. 2a). This may be similar to germination mechanics in fission yeast, where a polar cap stochastically wanders as the spore grows and ultimately breaks out once a critical strain is passed[32]. Morphodynamics are diffusion-dominated until the germ tube begins to bend, at which point deterministic forces begin to drive trajectories apart. Compound A introduces a plateau in the region where tip bending begins, revealing growth stunting, and also opens up new features with numerous bends. Extreme cases of heteregeneous growth, e.g., the difference between fungi with dramatically slowed growth and

others with completely unhindered growth, are captured by the Fokker–Planck model through plateaus followed by abnormally high gradients, shown in dark red. Compound B opens up a new branching feature region immediately following germination. Some fungi do develop normally, but without significant bending and with reduced growth rates. Compound X canalizes trajectories through only a subset of the features expressed in DMSO, namely straightening, with fungi closely following the landscape, before stunting occurs at a common location for most. Supplementary Figs. 1 and 2 show that Compound C at 0.041 mg L$^{-1}$ inhibits growth slightly, primarily through reduced diffusion rather than reduced landscape gradients, and increasing the concentration to 10 mg L$^{-1}$ drastically increases the stability of the spore region.

The data PDFs compare well with PDFs generated by a KDE of simulated trajectories for all conditions, validating the solution accuracy (Supplementary Fig. 3). Images from the videos can also be embedded in the morphospace and the mean-squared displacements of the video trajectories with those of the simulations show good agreement (Supplementary Fig. 4b, c). Furthermore, the entropy of the simulations always increases (Supplementary Fig. 4d). To get a measure of the uncertainty in the inferred landscapes and diffusivities, we trained the PINN three times for each compound, which exposed the algorithm to different data mini-batches. For each condition, the total losses decreased at similar rates (meaning the solutions are equally good at each training time), with overfitting occurring at approximately the same point. We therefore quantified uncertainty through the SD of the fields across repeats at the early stopping times described above (Supplementary Fig. 1b–f). The landscapes therefore provide intuitive comparisons of morphodynamics across conditions.

**Landscape deformations are caused by perturbations in the tip growth machinery.** The morphospace can also be used for data-driven development of a minimal mechanistic model to reveal potential causes of the landscape deformations discovered above. Having a single model with condition-dependent parameters provides further cell-mechanical characterization. We developed such a model for tip growth under all but Compound B and Compound C at 10 mg L$^{-1}$ (we excluded these because Compound B induces branching, which cannot be captured under the proposed model, and Compound C at 10 mg L$^{-1}$ does not permit a significant germ tube to develop). The morphospace reveals final shape and associated lengthening to be the primary degrees of freedom across the populations, motivating equations for length, $L$, and tip bending. From the videos, we observed three core features of lengthening (Supplementary Fig. 5a) as follows: that germination time is variable, that length increases approximately linearly with time, and that this growth rate is variable. We therefore modeled germination time, $t_g$, and subsequent linear growth rate, $\alpha$, as lognormally distributed, with growth rate distributed according to a reversed lognormal distribution truncated at zero. The lognormal distribution is widely used to model skewed phenomena across biology, including heterogeneous sensitivity to fungicides[33]. We compared three models for tip bending as follows: a simple random walk in growth direction, $\theta_\mathrm{global}$ (Fig. 4a); a random walk in the curvature of the growth path, $\kappa$; and a persistent random walk in $\kappa$ with parameters for stochasticity, $\sigma$, and relaxation to straight growth, $\tau^{-1}$ (see "Methods" for the mathematical expression of each model). Dynamics in $\kappa$ are a simple way to capture the effects of a diffusing growth zone, as has been described in fission yeast, where it is directed by landmark proteins concentrated at microtubule end points[34]. This connection can be made slightly more explicit

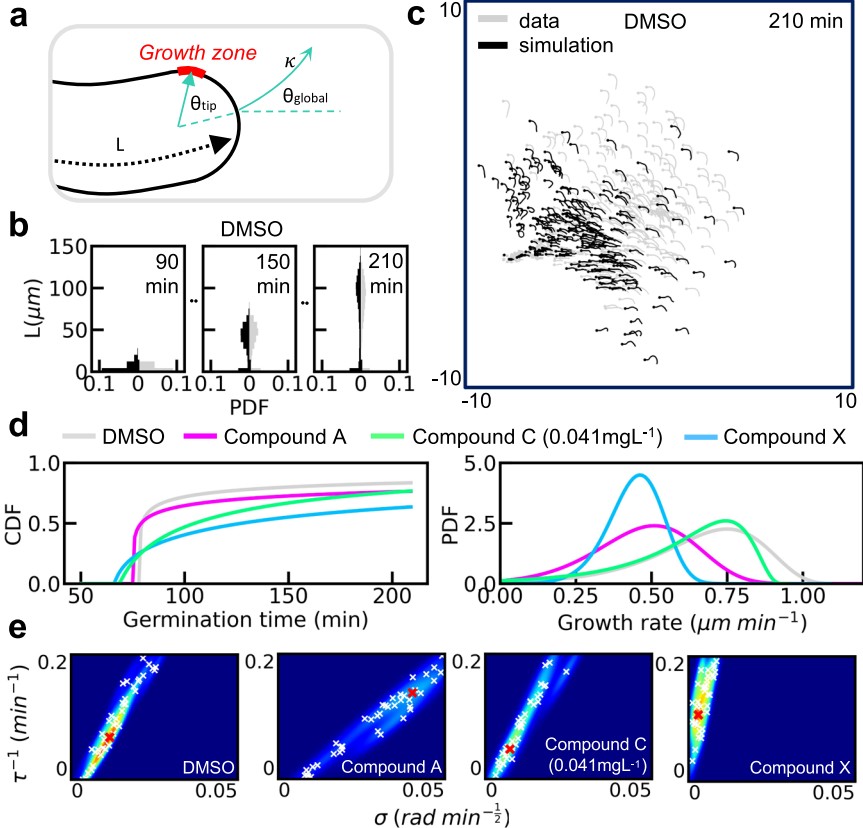

**Fig. 4 A persistent random walk model of the growth zone is fitted to image data. a** Tip growth is described with variables for length, L, linearly increasing in time, and path curvature, $\kappa$, which undergoes a persistent random walk, with relaxation to straight growth (i.e., a central growth zone). Dynamics of $\kappa$ may be the result of a diffusing growth zone, shown with angular location $\theta_{tip}$, which causes a changing direction of growth, described by $\theta_{global}$ in the lab frame. **b** All parameters were fitted using approximate Bayesian computation with sequential Monte Carlo (ABC-SMC). Lengthening parameters were fitted using length (L) histograms at nine equally spaced time points (three of the nine DMSO snapshots are shown here, with data in gray and simulations in black), between 90 and 210 min after mixing with solution. **c** Bending parameters were fitted by comparing morphospace embeddings of the 210 min snapshot data with those of images simulated with MAP lengthening parameters, as shown here for DMSO. **d** MAP germination time cumulative density functions (CDFs) and growth rate probability density functions (PDFs) show typical perturbations include premature germination, reduced germination frequency, and reduced maximum and mean growth rates. **e** Bending parameter posteriors (for stochasticity, $\sigma$, and relaxation to straight growth, $\tau^{-1}$) show final morphologies depend primarily on the ratio of the two bending parameters and fungicides can both increase and decrease this ratio. Accepted parameters of the final ABC-SMC population are plotted in white, with MAP values in red. Source data are provided for **d**, **e**.

by the introduction of an angular growth zone position, $\theta_{tip}$, and a mapping $\kappa = f(\theta_{tip})$. Although $f$ is unknown, this function is likely monotonically increasing and passing through the origin (i.e., a central growth zone corresponds to straight growth). Supplementary Fig. 5b shows $\theta_{global}$ variation in time for the three models, for some intuition on the dynamics.

For both parameter inference and model selection, we used ABC with sequential Monte Carlo (ABC-SMC, details on the SMC are in "Methods")[35]. In ABC, parameters are selected from a prior distribution and simulations are run. If the simulations are within some threshold of similarity with the data, then the parameters are stored. The density over the stored parameters forms the posterior distribution. For model selection, model index is introduced as an additional parameter and the posterior distribution is found over the joint space of model index and parameters, with the marginal distribution over model index giving the model probabilities. This biases towards low-dimensional parameter spaces, favoring the most parsimonious models. For the parameters involved in lengthening, we used histograms of fungus lengths at each snapshot (see Fig. 4b for three of the nine DMSO snapshots). For the bending parameters, we then simulated fungus images using maximum a posteriori probability (MAP) lengthening parameters and compared

histograms of their morphospace embeddings with those of the 210 min snapshot data (Fig. 4c for DMSO). Further details can be found in "Methods." Compound A data were used for model selection, as it covered the full spectrum of features. Only Model 3 could reproduce the data with high accuracy (in particular the multiple bends, where relaxation to central growth is required to reproduce the alternating bending direction). This match is shown by the probabilities of the three models and MAP simulations (Supplementary Fig. 5c, d). The equations governing dynamics after germination are therefore

$$\text{for } t > t_g \begin{cases} dL = \alpha dt \\ d\kappa = -\tau^{-1}\kappa dt + \sigma dW \end{cases}, \quad (5)$$

where dW is the Wiener process, and for $t \leq t_g$, we have $L = \kappa = 0$.

Figure 4d shows the cumulative density functions of germination time and PDFs of growth rate associated with MAP parameters for each condition. Germination time is strongly skewed, with fungicides inducing premature germination and reducing subsequent germination frequency. Growth rates are less skewed, and both the maximum and mean growth rates are reduced by all fungicides. The bending posterior distributions (Fig. 4e) have linear shape, showing that for all conditions, the

morphology depends primarily on the ratio of $\tau^{-1}$ to $\sigma$. Compounds A and X induce decreased and increased ratios of relaxation to stochasticity, respectively. Comparisons of MAP simulations with data for all conditions are shown in full in Supplementary Fig. 6.

## Discussion

Phenotypic screens are often used to identify drug efficacy and mode of action by comparing visible features such as morphology. However, such screens are often limited to human-defined and static features, and any incorporation of dynamics typically focuses on stereotyped behaviors. Here we characterized morphodynamics of the Asian soybean rust crop pathogen, *P. pachyrhizi*, germinating in vitro in the presence of different fungicides, directly from image sets. We found that morphodynamics are diffusion-dominated until the tip begins to bend, at which point deterministic forces begin to drive trajectories apart. Fungicide-induced landscape deformations include barriers, plateaus, and canalized pathways. These features may arise from physical perturbations including premature but lower-frequency germination, reduced growth rates, and both increased and reduced stabilities of the growth zone. The global morphospace therefore allowed us to extract meaningful morphodynamic parameters directly from images, revealing perturbed driving forces in the Fokker–Planck model, and providing a similarity metric for the tip growth model. For both models, the nonlinear embedding affords crucial interpretability through visualization. Moreover, the two models give complementary views of the dynamics. Taking Compound X as an example, the landscape model reveals low diffusion following germination when compared with DMSO. Hence, once germinated with Compound X, fungi grow at very similar rates, with little bending, tightly following the underlying landscape. The tip growth model similarly shows a narrowed distribution of growth rates and reduced bending, and these observations are confirmed when reviewing the time-lapse videos after the analysis.

Despite many benefits, the analysis in its current form has limitations. For systems with higher-dimensional dynamics, a 2D morphospace may be unsuitable. For instance, we omitted a compound that induced blistering along the germ tube, which expanded the compound's morphospace beyond two dimensions. For such cases, the bottom-up mechanistic model parameter fitting can be done in higher dimensions but at the cost of increased computation time and reduced interpretability. For the Fokker–Planck model, higher-dimensional systems may still be characterized in terms of networks of attractors[18] and the marginal dynamics of pairs of morphological degrees of freedom could still be visualized in 2D. Such characterizing of dynamics over nonlinear representations may be most powerful when different conditions cover similar features, such that differences have a physical rather than algorithmic origin. As well as being able to characterize dynamics across multiple conditions, these simple models are useful for systems without a priori established dynamics.

Do the landscapes correspond to any physical quantities beyond representing a distilled statistical representation? The potentials represent the deterministic part of the motion, e.g., extension of the germ tube from turgor pressure and vesicle delivery (the potentials largely drive in the direction of increasing length). Furthermore, diffusion captures features that vary at single-fungus level, including the precise growth rate, and the direction of bending (potentially caused by a diffusing growth zone or noise[34]). The observed phenotypes are certainly plausible based on putative modes of action of the drugs in terms of inhibiting microtubules, kinases, and gene expression (see Supplementary Note 4).

An interesting area for future work is to extend unsupervised morphodynamic analysis beyond minimal characterizations, to more detailed models. This could be achieved by joint learning of the underlying representation and equations of motion, e.g., by minimizing the prediction error and complexity of the equations of motion. Such approaches have been shown capable of recovering physical laws in Cartesian coordinates from warped video footage[36] and it would be interesting to extend this to complex biological systems, where the underlying laws are less clear. Modeling of cell growth in terms of generalized shape coordinates is an area of active research, with one promising model balancing dissipative, mechanical, and active forces[37]. Another interesting area for future work is to better understand the connection between internal mechanics and morphodynamics. This could be achieved by joint modeling morphology and organelles (using flourescent markers), conditional on various pharmacophores. Interesting organelles and molecular processes may include secretory vesicles for membrane delivery, small GTPases for growth cone labeling, and motor and cytoskeletal proteins for transport and structure[38]. Interpretable representations for each could be found using nonlinear dimensionality reduction, as done here for morphology[39].

Although often hailed as the future of deep learning, use of unsupervised learning techniques within the natural sciences often stops at low-dimensional data visualizations. We hope the work presented here may stimulate further work leveraging the discovery power of unsupervised methods within interpretable physical models.

## Methods

Algorithms were run in Python and packages used are detailed in Supplementary Methods.

**Imaging and image processing**. For the snapshot data, spores were mixed in each of the six treatment solutions and imaged in 96-well plates on the Opera QEHS running Opera Software 2.0 (EvoShell, Opera CHKN/QEHS Red Ver. 2.0.0.12017 Rev.: 89046, PerkinElmer, Inc.). At nine equally spaced times between 90 and 210 min, cell walls were stained with Calcofluor White solution with KOH, which fixated the fungi. Each snapshot was therefore of a different batch of spores. The staining procedure enabled the collection of two images for each well and time point, using different excitation wavelengths, one showing the spores and another showing the germ tubes. For the time-lapse videos, images were taken at 3 min intervals on the JuLI Stage Real Time Cell History Recorder running JuLI Stage V. 2.0.1 and JuLI EDIT V. 1.0.0.0 (NanoEnTek, Inc.), without staining, meaning only one image was collected for each well and time, showing the full fungi. All imaging was done at ×10 magnification. Full details on the imaging can be found in Supplementary Information. The snapshots and time-lapse videos were processed differently, because (a) the snapshots had spores and germ tubes separated, which we took advantage of to automate alignment, and (b) the small number of time-lapse videos meant we could manually align these for higher precision.

For processing the snapshot images, we used adaptive binarization (to account for lighting defects) to get two images for each view: one of germ tube contours, another of spore contours. Adding these together then gave an image with full fungi. Contours with an area above a threshold found by trial and error were removed as obvious overlapping fungi and the remaining ones were cropped by finding the minimum bounding rectangle. These regions of interest were rotated to align with the pixel grid and padded so all were 200 × 200 pixels, to fit the largest fungi in the set. Incomplete fungi were also removed at the image borders. We then used a supervised convolutional network, trained on a sample of hand-labeled images, to remove contours that contained overlapping fungi. Remaining individual fungi were then translated and rotated so the initial growth directions coincided, and a flip was executed if the right-most point of the fungus was higher than the germination point (see Supplementary Note 1). Finally, we replaced all spores with identical circles, so as to prioritize modeling of the germ tube; the resulting morphospace point is then widened into a spore region through the KDE.

For the time-lapse videos, we again binarized the frames (non-adaptive this time, as there were not significant lighting defects). We then found a series of contours across frames, whose centers of mass were closest, and manually looked through these series to find those that corresponded to tracking an individual fungus. We then manually aligned these so that the initial growth directions coincided, this time using ImageJ and Gimp. Before being inputted into the autoencoder, snapshot and time-lapse video pixels were assigned to a value in the

set {0, 1}. Full details on the image processing can be found in Supplementary Note 1.

**Neural networks**. For the autoencoder's encoder, we used four convolutional layers with 16, 32, 64, and 16 feature maps, all with $3 \times 3$ kernels, Rectified Linear Unit (ReLU) activations, batch normalization, and alternating stride sizes of 1 and 2 in PyTorch. The decoder's structure mirrored the encoder's, but with transposed convolutions. We used a sigmoid output activation and binary cross entropy loss, over mini-batches of 50 images, and trained for 4 epochs using the Adam optimizer[40] with a learning rate of $10^{-4}$, which took 2 h with a Quadro RTX 6000 GPU card. Training was stopped at the point at which the trajectories of the single-fungus videos were least complex.

For the PINN, the loss function to be minimized comprises four terms, with the first three calculated over random mini-batches of $N$ data points and the final one over the full spatial grid of $M$ data points. The first is the mean-squared difference between the learned PDF, $\hat{p}(\mathbf{x}^j, t^j)$, and data $p(\mathbf{x}^j, t^j)$,

$$L_{PDF} = \frac{1}{N} \sum_{j=1}^{N} [\hat{p}(\mathbf{x}^j, t^j) - p(\mathbf{x}^j, t^j)]^2, \tag{6}$$

with $\{\mathbf{x}^j, t^j\}$ in the nine snapshots. The second is the mean-squared PDF at the boundary,

$$L_{BC} = \frac{1}{N} \sum_{j=1}^{N} [\hat{p}(\mathbf{x}^j, t^j)]^2 \tag{7}$$

with $\{\mathbf{x}^j, t^j\}$ selected from $10^6$ uniformly distributed boundary points, and the third term is the mean-squared PDE residual ($\mathcal{N}$, given in Eq. (3)),

$$L_{PDE} = \frac{1}{N} \sum_{j=1}^{N} [\mathcal{N}(\hat{p}(\mathbf{x}^j, t^j), \hat{D}(\mathbf{x}^j, t^j), \hat{U}(\mathbf{x}^j))]^2, \tag{8}$$

with $\{\mathbf{x}^j, t^j\}$ selected from $10^6$ points uniformly distributed over the whole domain. The final term ensures the PDF integrates to one:

$$L_{norm} = \left[ \sum_{j=1}^{M} \Delta x_1 \Delta x_2 \hat{p}(\mathbf{x}^j, t) - 1 \right]^2, \tag{9}$$

with $\mathbf{x}^j$ covering the full spatial grid and $t$ randomly selected. For the total loss (Eq. (4)), we used hyperparameters of 1, 1, 500, 0.01 for $a$, $b$, $c$, and $d$, with reasons discussed in Supplementary Note 2.

The three PINN neural networks had 5 fully connected layers, each with 50 neurons, with residual skip connections, and swish activations between layers. Output variables that share inputs (e.g., the PDF and diffusivity) can be outputted from a single neural network if they are likely to have similar features, for increased computational efficiency. We used the Adam optimizer[40] with a learning rate of $5 \times 10^{-4}$ and batch sizes, $N$, of 8000. To speed up training, the DMSO landscape was first trained for 10 h and PINNs for the other conditions were initialized with these weights (known as transfer learning). For forward simulations over the landscapes, particle starting positions were sampled from the initial PDF learned by the PINN and then simulations were run by evaluating the potential and diffusivity on a $1000 \times 1000$ spatial grid, with 20 snapshots in time for the diffusivity, and simulating Eq. (1) with a time step of 0.01 min.

**Tip growth model**. The three-parameter lognormal PDF is given by

$$f(x; s, \sigma^2, \text{loc}) = \frac{1}{\sigma \sqrt{2\pi}(x - \text{loc})} \exp \frac{\log^2 \left( \frac{x - \text{loc}}{s} \right)}{2\sigma^2} \tag{10}$$

where $\sigma$ is a shape parameter, $s$ is a scale parameter (also the median), and loc is a location parameter (the lower bound). The two-parameter distribution has loc set to zero.

We modeled germination time, $t_g$, as distributed according to $t_g \sim \text{lognormal}(s_{t_g}, \sigma_{t_g}, \text{loc}_{t_g})$, and growth rate, $\alpha$, as distributed according to $\alpha = \text{loc}_\alpha - x$, with $x \sim \text{lognormal}(s_\alpha, \sigma_\alpha, 0)$ and resampling for negative $\alpha$. Length data were extracted by summing the binarized fungus images, and both the lengthening and bending parameters were fitted using ABC-SMC[35]. This is a computationally efficient implementation of ABC, identifying intermediate distributions over a series of populations, and gradually decreasing the acceptance threshold. All histograms were compared using the summed absolute distance and we trained the autoencoder for an extra two epochs with simulations generated randomly from the prior distribution to get coverage of any novel features.

We compared three models for tip bending, with Model 3 found to reproduce the data best. For all of the following, $\sigma$ is a noise parameter that was fitted and $dW$ is the Wiener process. Figure 4a shows a schematic with the bending angles and curvature. Model 1 was a random walk in the global direction, $\theta_{global}$, a simple model commonly used in the literature:

$$d\theta_{global} = \sigma dW. \tag{11}$$

Model 2 was a random walk in the curvature of the growth path, $\kappa$, in order to connect to cell tip mechanics:

$$d\kappa = \sigma dW. \tag{12}$$

Model 3 was a persistent random walk in the curvature, with an additional parameter, $\tau^{-1}$, for relaxation to straight growth, motivated by work analyzing fission yeast tip growth mechanics[34]:

$$d\kappa = -\tau^{-1} \kappa dt + \sigma dW. \tag{13}$$

See Supplementary Note 3 for details on the creation of the simulation images and settings used for running ABC-SMC.

**Reporting summary**. Further information on research design is available in the Nature Research Reporting Summary linked to this article.

## Data availability
The image data that support the findings of this study have been deposited at http://cellimagelibrary.org/groups/54615. The figure data generated in this study are provided in the Supplementary Information. Source data are provided with this paper.

## Code availability
The code used, along with a subset of images, are available at https://github.com/hcbiophys/morphodynamics[41].

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

## Acknowledgements

We thank Suhail Islam for invaluable computational suppport. This work was funded by the Biotechnology and Biological Sciences Research Council (grant number BB/M011178/1) and Syngenta provided financial and technical support in the form of an iCASE studentship to H.C.

## Author contributions

H.C. and R.G.E. designed and H.C. performed the theoretical analysis. A.M. and G.S. contributed the reagents and performed the imaging. H.C. and R.L. did the image processing. All authors wrote the paper.

## Competing interests

The authors declare no competing interests.
