## [Peer Review File · Nature Communications]

Physics-informed deep learning characterizes morphodynamics of Asian soybean rust diseaseReviewers' Comments:

Reviewer #1:

Remarks to the Author:

The authors present an approach of combining unsupervised deep learning with biophysical modelling to characterize and study shape changes (morphodynamics) for cell-drug interactions. This approach is applied to the studies of the development of the Asian soybean rust crop pathogen. There are two approaches introduced in this context: top-down approach of learning-based morphodynamic landscape studies and bottom-up approach of biophysical modeling/simulations of tip growth. The ultimate goal of such studies is drug efficacy assessment.

In general, I find this concept interesting, sufficiently novel and in line with the latest trends of automatic extraction of both spatial and temporal features from images/videos using machine learning and creating morphodynamic multi-dimensional latent spaces, ideally reducing the number of dimensions of such spaces and trying to make the dimensions and movement inside them explainable. Thus, I find this work worth publishing but after addressing some issues:

Major issues:

1) It is quite difficult to follow and understand the methodology because the description of methods is distributed into multiple places of the paper – quite large part is included in the Results section (I do not understand why, Results should not contain methods, not speaking about equations, in my opinion), small part in the Methods section and the rest in the Supplementary Material. None of these parts contains complete description, not speaking about complete definitions of all symbols/parameters. Often, symbols are not defined and one must guess what they mean. For example, what is the sense of section 4.3? It contains just one paragraph starting with a sentence with many undefined symbols (some of them mentioned earlier in the paper but some not defined even earlier). Similarly, section 4.2 is extremely short and just referencing twice the Supplementary Information and also methods/equations/symbols introduced elsewhere. I would expect to find complete methodology in the Supplement, short summary in Methods in the main paper (but avoiding undefined symbols) and no methodology in Results.

2) The Imaging and Image Processing part (section 4.1 and Supplement section 3) is extremely short and rather unclear.

2A) Apparently, you used two different inputs: live time lapse videos with 3 min sampling interval and a series of 9 time snapshots after fixation and staining with 15 min interval between 90 and 210 min after mixing with solution, i.e. different batch of spores for each time point. I understood that both types of inputs (3 min as well as 15 min) were performed after mixing with different compounds (DMSO, compounds A, B, C, X). But it is not stated why it was necessary to have 2 kinds of inputs, why wasn't it enough to image just 3 min live videos (that have 5 times better sampling interval) like those shown in the supplement materials? They look quite well, have a good contrast, easy to observe tips growing. It would help to add characteristics of both types and some pros/cons and substantiation why you chose 9 time points with 15 min interval for the fixation case. It would also help to add some examples of 15 min input sequences. Also, the number of final sequences/videos and number of final ROIs is missing, I found the number of ROIs only in Reporting Summary Document, which says 700,000 single-fungus images. But how many of them was from 3 min videos and how many from 15 min sequences?

2B) You state that final images were 200x200 pixels but how did you find these ROIs? Have you performed some resampling? In Supplement, you speak about fully-automated processing. Then you should describe the algorithm. You mention only "adaptive binarization", which is a vague term and says nothing about further binary/thresholded image processing to find and crop objects, not speaking about dealing with overlapping or incomplete objects. Also, you aligned the ROIs based on initial

growth direction - again, which algorithm did you use? For 3 min videos, you speak about manual alignment in Gimp, which is in contradiction with fully-automated processing. Did you use automation just for 15 min sequences and manual approach for 3 min? Please state U-Net parameters and configuration. Why was U-Net used only for 3 min videos, not 15 min sequences? Finally, you say you binarized the images to [0, 1] interval. This does not make sense at all - either you rescaled the values to [0, 1] interval creating a float image or you binarized the image to {0, 1} set of values and created a binary image with just two values for each pixel. Anyway, it seems the methodology was different for 3 min and 15 min input sequences, so it would be best to have a separate description for each of these two types.

3) The selection of compounds is not discussed. Why have you chosen the given compounds? Also, you mention (not in the paper but in the Reporting Summary) that you excluded one of the compounds because your approach could not cope with it. If this was the case, it would be fair to admit it in the paper. Furthermore, you excluded compounds B and C@10mgL⁻¹ in Section 2.3 (tip growth modeling) - why? Were they too complicated to model?

Minor issues:

1) Fig. 3 and Fig. S1: White circles for 90 min in the legend are probably a mistake - there are no such circles in the figures, instead there are pink circles and white temporal paths interconnecting pink and cyan circles.

2) Fig. S1c: why do all real data plots end at approx. time 160 min? What happened between 160 min and 210 min? Was "good agreement" of real and simulated plots assessed only using naked eye or somehow mathematically after taking into account the temperature-induced time shift?

3) Section 4.3 and Suppl 6: you have plenty of undefined symbols here (some were mentioned elsewhere like α and t_g , some not). I guess sigmas are square roots of variances and s stand for means, right? What is loc ? Usually, lognormal distributions have two parameters - mean and variance, you have three. Also, variance parameter is usually denoted as sigma squared. Substantiation of using lognormal distribution is missing.

4) Section 4.3 and Suppl 6: It would be nice to include some examples of the simulation images.

5) Figure 4: Angle θ_{global} is missing in the legend. Vice versa, L is missing in the drawing - it would be nice to add - I guess it is just length from the left side of the image along medial axis of the tip, right?

6) Eq. 9: You defined two cases: $t > t_g$ and $t < t_g$. The case $t = t_g$ is missing.

Reviewer #2:

Remarks to the Author:

Cavanagh et al. presented a physics-informed deep-learning method to characterise the in-vitro morphological changes of the Asian soybean rust crop pathogen, *P. pachyrhizi*. Specifically, the authors applied an autoencoder model to extract salient morphological features of the single-fungus images (under different perturbation conditions), and thus created a 2D "morphospace" embedding for visualisation. They then used a Fokker Planck model to interpret the morphodynamic driving forces in the form of Waddington landscapes. Also, in combination of a random-walk-based model of the tip-growth that is fitted to the image dataset, the phenotype transitions during the growth process can be inferred.

This work aligns well with the emergent field of cellular morphological phenotyping (especially with the

help of computer vision), which is gaining popularity in a wide range of disciplines, cell biology, plant biology and microbiology etc. The methodology presented in this work is generally rigorous. The results are technically sound, and demonstrated the feasibility of this integration of unsupervised learning and biophysical modelling in characterising the Asian Soybean Rust Disease, especially offering some insights into the morphological dynamics during growth - which is I believe the main contribution of the work.

Having said that, it is not clear to me, in the current form of this manuscript, how this physics-informed neural network (PINN) phenotyping strategy is superior than the existing approaches (i.e. can other methods reveal similar observations? What are their limitations?). Furthermore, how does this work (especially the phenotyping pipeline) can be widely applicable to large phenotypic screens of other agricultural biocides. I believe these are the key issues that the authors might want to consider to articulate in order to ensure the work is suitable for the wide readership of Nature Communications. Apart from that, here below are some further technical questions/comments:

1. Regarding the use of autoencoder for salient feature extraction, what is the impact (is it sensitive?) on the pipeline performance if the architecture of the autoencoder is modified? Or even another type of CNN model is used for unsupervised feature extraction?
2. It is mentioned that the diffusion over the landscapes has two sources: morphodynamic diffusion, and embedding diffusion. It is rather unclear how to distinguish and validate the two factors.
3. In terms of visual interpretation, I found it's rather non-trivial to relate (map) the information between the morphodynamic landscapes (e.g. Fig. S1a) and Morphospace features extracted from the decoder (e.g. Fig. S1b). It would be helpful if the authors offer a clearer visual guide in terms of the graph representation in the figure.
4. To enable the inference of landscapes, the authors used the snapshot embeddings to infer the driving forces through a Fokker Planck model. In the model, the PINN loss (Eq. 8) is defined:
 - The authors stated that "lower frequency functions are explored first". I wonder if this approach was rigorously validated in this study (although it's common to argue with "Occam's razor").
 - Regarding the set of hyperparameters of a , b , c and d , how sensitive is the entire pipeline performance to the change of values of these parameters?

Reviewer #3:

Remarks to the Author:

Please see attached PDF document for review.

Reviewer #4:

Remarks to the Author:

This paper shows, for the first time, a characterization of phenotypic screens, i.e. morphodynamics, of Asian soybean rust germinating in presence of different fungicides based on image sets using unsupervised methods within interpretable physical models. The method replaces human categorisation of complex phenotypes. It allows to characterizing morphodynamics (diffusion versus forced) and infers fungicide-dependent landscapes using Fokker-Planck-PDE and highlights forward simulations over the landscapes. The combined approach is highly novel and very convincing (advanced methods of deep learning, applied to important disease x fungicide-complex). I think that the knowledge gained by the proposed method will inspire further research in the highly challenging research area of deep learning and will allow further applications (e.g., in smart soil image analysis or probably systems with more than two players).

Yet, I do have two (probably minor) concerns:

The first relates to the choice of the fungicides, i.e. to "compound X". I am just wondering if this

“syngenta research compound” would be a) freely available for future research, and, in particular b) allow supporting the reproducibility of the study - is there really no conflict of interests? The second point relates to the model choice and further elaborations on the tip-bending model (l. 163). As far as I understood, compound A data was used for model selection (Sup Fig 3b + lines 184/185). However, the further results refer particularly to compound A data, e.g. the authors claim that “benefit of model 3 (for relaxation) is most clear for compound A” (l. 196). Well, I think this is not a good point, since model 3 was selected just because of being best performer for compound A.

Minor remarks:

- Last paragraph in introduction: I expected to find the goals of the study (rather than the achievements, e.g. 52, 54).
- Fig 1a: Visualisation on top of the image snapshots in 1a: Maybe indicate what to see here?
- l. 72 /73: I am not clear about the difference between compound- and condition-dependent. What are “conditions”? See also figure caption for Fig 3 “for DMSO, and the white outline shows the same but for the other conditions”.
- Fig 2a: I think it might be easier to grasp the idea of human versus autoencoder if this part of the figure would have been split in to a/b. What is the difference between “inset” in 2a (l 73) and “indent” in 2a (figure caption)? Probably indicate the adding value of small graph with the “lower dimensional manifolds”.
- Fig 2: Is there a reason for indicating the spores in 2a and 2b differently (in 2b with arrow)? The figure caption should probably include details on the dashed arrows for capturing lengthening of spores and variation in shape.
- Fig 3d: Are the particles on the landscapes at $t = 90$ min coloured in pink (not white)? But, I may be wrong and the tiny sphere are not representing the particles?
- Eq1 vs Eq2: The last term in each equation is commutative ($D_p = pD$). Thus, why change the order?
- Eq4 – Eq6: Assuming that each of these three parts of the total loss function are calculated over random minibatches (= sample), why divide by N and not by N-1?
- L. 132: Probably, should add a reference for “precluding modelling on representations found with unsupervised learning”?
- L. 134: Would it be helpful to learn more about the “threshold energy required for germination” – was not mentioned before.
- L 133/136: Suppl. 1a shows data for compound C, not explicitly for DSMO (yet it is mentioned there that both have large similarities).
- L. 149: explain abbreviation “KDE”, e.g. to fig caption of Fig. 3
- L. 181: Fi4b: “for a subset of DSMO snapshots”. Why a subset? Dimension?
- Fig 4b: unclear about the colour code: what is grey/black and data of which compound are shown here?
- Fig 4c: The main text (l. 183) indicates it is DMSO data and not data of other compounds

Response to the Reviewers

Thank you very much for taking the time to provide such useful comments, which we feel greatly improved the manuscript. Here, we address the issues raised, and indicate the changes made (in blue) to the manuscript and the supplementary information file.

Response to Reviewer #1 (Expertise: computer science, image processing):

The authors present an approach of combining unsupervised deep learning with biophysical modelling to characterize and study shape changes (morphodynamics) for cell-drug interactions. This approach is applied to the studies of the development of the Asian soybean rust crop pathogen. There are two approaches introduced in this context: top-down approach of learning-based morphodynamic landscape studies and bottom-up approach of biophysical modeling/simulations of tip growth. The ultimate goal of such studies is drug efficacy assessment.

In general, I find this concept interesting, sufficiently novel and in line with the latest trends of automatic extraction of both spatial and temporal features from images/videos using machine learning and creating morphodynamic multi-dimensional latent spaces, ideally reducing the number of dimensions of such spaces and trying to make the dimensions and movement inside them explainable. Thus, I find this work worth publishing but after addressing some issues:

Major issues:

1) It is quite difficult to follow and understand the methodology because the description of methods is distributed into multiple places of the paper – quite large part is included in the Results section (I do not understand why, Results should not contain methods, not speaking about equations, in my opinion), small part in the Methods section and the rest in the Supplementary Material. None of these parts contains complete description, not speaking about complete definitions of all symbols/parameters. Often, symbols are not defined and one must guess what they mean. For example, what is the sense of section 4.3? It contains just one paragraph starting with a sentence with many undefined symbols (some of them mentioned earlier in the paper but some not defined even earlier). Similarly, section 4.2 is extremely short and just referencing twice the Supplementary Information and also methods/equations/symbols introduced elsewhere. I would expect to find complete methodology in the Supplement, short summary in Methods in the main paper (but avoiding undefined symbols) and no methodology in Results.

Thank you for bringing to our attention that the methods are spread out throughout different parts of the manuscript and supplementary information. We absolutely agree on the importance of having information self-contained and readily interpretable. We have therefore made the following changes to the structure:

- Equations 4-7, describing in detail the components of the physics-informed neural network (PINN) loss terms, have been moved to Methods, and Results now contains only a high-level summary.
- For Section 4.2, we now have the details of the PINN loss terms, as well as more information on the structure and hyperparameters of the neural networks. Supplementary Information (Section S4) now contains all of this information repeated (so as to be complete and self-contained), as well as further details and discussion points.

- For Section 4.3, we now start with an explanation of the terms parameterizing the lognormal distribution for context of the tip growth parameters.

2) The Imaging and Image Processing part (section 4.1 and Supplement section 3) is extremely short and rather unclear.

Thank you for bringing our attention to this point. We feel that the contributions of the paper are in the morphodynamic analysis after the data pre-processing, which is specific to our data sets, and not necessarily the only viable approach. However, we acknowledge that it will be useful for readers to have the complete pre-processing methodology to get an idea of the options available to them when carrying out similar object extraction on their image sets.

We have therefore now expanded the description in Supplementary Information (Section S3), with a summary of the main points now in Methods.

2A) Apparently, you used two different inputs: live time lapse videos with 3 min sampling interval and a series of 9 time snapshots after fixation and staining with 15 min interval between 90 and 210 min after mixing with solution, i.e. different batch of spores for each time point. I understood that both types of inputs (3 min as well as 15 min) were performed after mixing with different compounds (DMSO, compounds A, B, C, X). But it is not stated why it was necessary to have 2 kinds of inputs, why wasn't it enough to image just 3 min live videos (that have 5 times better sampling interval) like those shown in the supplement materials? They look quite well, have a good contrast, easy to observe tips growing. It would help to add characteristics of both types and some pros/cons and substantiation why you chose 9 time points with 15 min interval for the fixation case. It would also help to add some examples of 15 min input sequences.

Thank you for suggesting improved clarity with regards to motivating the two different types of image data.

In practice, such morphodynamic phenotyping screens are planned to be extended to hundreds of compounds with independent snapshot images. The time-lapse videos are much harder to acquire (even with the alternative JuLI Stage imaging device), and it is not possible to do this with more wells in parallel. In short, it is unsuited to high-throughput screening needed for statistically relevant results, and also harder to maintain a constant temperature than with the snapshot assays. Hence, they are not used in industry. Finally, and a minor point: the time lapse images are harder to align fully unsupervised than the snapshot images, because the fixation with Calcofluor White staining solution allowed separate images showing spores and germ tubes for each view.

We therefore used the small number of time-lapse videos to validate the inferred Fokker-Planck parameters, and to motivate the model form for the tip-growth model, and used the statistically significant high-throughput snapshot assays for inferring parameters. 9 time points with 15 min intervals were chosen for the snapshot images to be numerous enough to capture most of the interesting growth features immediately following germination, within a feasible time frame for preparing the assays. We discuss in Section S4 how the inference becomes more accurate with more snapshots, however.

We have explained more clearly the reason why the two different image types were used, and how they were utilised during the analysis in Section 2.1 (after introducing the compounds). We have also added an example time-lapse sequence (Fig. S4a).

Also, the number of final sequences/videos and number of final ROIs is missing, I found the number of ROIs only in Reporting Summary Document, which says 700,000 single-fungus

images. But how many of them was from 3 min videos and how many from 15 min sequences?

Thank you for raising this point. This was a mistake and we have now corrected it to detail that there were 600,000 single-fungus snapshot images (mean 11,000, standard deviation 3,000) and have specified how many time-lapse videos there were across compounds (3-8). This information is now detailed in Section 2.1 after introducing the compounds.

2B) You state that final images were 200x200 pixels but how did you find these ROIs? Have you performed some resampling? In Supplement, you speak about fully-automated processing. Then you should describe the algorithm. You mention only “adaptive binarization”, which is a vague term and says nothing about further binary/thresholded image processing to find and crop objects, not speaking about dealing with overlapping or incomplete objects. Also, you aligned the ROIs based on initial growth direction - again, which algorithm did you use? For 3 min videos, you speak about manual alignment in Gimp, which is in contradiction with fully-automated processing. Did you use automation just for 15 min sequences and manual approach for 3 min? Please state U-Net parameters and configuration. Why was U-Net used only for 3 min videos, not 15 min sequences? Finally, you say you binarized the images to [0, 1] interval. This does not make sense at all – either you rescaled the values to [0,1] interval creating a float image or you binarized the image to {0, 1} set of values and created a binary image with just two values for each pixel.

Thank you for suggesting the inclusion of more details on the image pre-processing. We have added a full detailed description now to Supplementary Information (Section S3), with the main points also in Methods (Section 4.1). Specifically for the points you raise:

- We have explained how ROIs were found by adaptive binarization for the snapshot images to account for large-scale lighting defects (now with the function and parameters used for this adaptive binarization detailed in Supplementary Information) and regular binarization was used for the time-lapse videos, as these do not have significant lighting defects.
- For aligning the ROIs based on initial growth direction, for the snapshot images we used the fact that we had two images for each view (one with spores highlighted, another with germ tubes highlighted) and that there was a slight overlap of these two. Section S3 now has full details of the alignment procedures for both image types, saying that snapshot pre-processing was automated, but the time-lapse pre-processing was manual.
- Our apologies for the confusion with U-Net. We had initially used this algorithm to automate processing of the time-lapse videos. However, for simplicity (since we only tracked a small number of fungi), we manually segmented the spores (using ImageJ) and subsequently aligned them using Gimp. This process is now detailed in Section S3.
- We have corrected the terminology to more clearly indicate that all pixel values were elements from the set {0, 1}.

Anyway, it seems the methodology was different for 3 min and 15 min input sequences, so it would be best to have a separate description for each of these two types.

We agree that there could be more clarity distinguishing the two different image types and processing used. We have therefore partitioned information on processing the snapshots and time-lapse videos into two distinct paragraphs for clarity (in both Section 4.1 and Section S3).

3) The selection of compounds is not discussed. Why have you chosen the given compounds?

Thank you for suggesting an explanation for how the compounds were initially selected, which we agree will improve the manuscript. These compounds were identified at pre-screening stage by eye to show qualitatively interesting (different to DMSO, depending on concentration) phenotypes. They were therefore selected as ideal compounds to test our proposed morphodynamic screening pipeline on, to see if it could intuitively differentiate morphodynamics. We have clarified this in Section 2.1 and Section S2.

Also, you mention (not in the paper but in the Reporting Summary) that you excluded one of the compounds because your approach could not cope with it. If this was the case, it would be fair to admit it in the paper.

Thank you for this suggestion. We agree that it is useful for readers to be aware both of the limitations of a 2D approach and exciting possibilities in extending the analysis to higher dimensions. We have added this point in the Discussion (Section 3).

Furthermore, you excluded compounds B and C@10mgL⁻¹ in Section 2.3 (tip growth modeling) – why? Were they too complicated to model?

We did not include Compound B and Compound C at 10mgL⁻¹ because Compound B induced branching phenotypes, which cannot be captured using the proposed minimal model, and Compound C at 10mgL⁻¹ does not permit a significant germ tube to develop. We have added this clarification to the manuscript (Section 2.3).

Minor issues:

1) Fig. 3 and Fig. S1: White circles for 90 min in the legend are probably a mistake – there are no such circles in the figures, instead there are pink circles and white temporal paths interconnecting pink and cyan circles.

Thank you for pointing this out - we have corrected the colouring description.

2) Fig. S1c: why do all real data plots end at approx. time 160 min? What happened between 160 min and 210 min? Was “good agreement” of real and simulated plots assessed only using naked eye or somehow mathematically after taking into account the temperature-induced time shift?

The video time-lapse images were taken at a higher temperature and they therefore reach approximately the same morphological state that the snapshot fungi do at 210 min, at ~160 min.

We had previously compared the simulation and data mean squared displacements by eye, but have now added a ‘confusion matrix’ to Supplementary Fig. S4c showing the mean absolute differences for each data-simulation pairing. The shown mean absolute difference for each pairing is the minimum found after shifting the time series horizontally, to account for the temperature-induced shift.

3) Section 4.3 and Suppl 6: you have plenty of undefined symbols here (some were mentioned elsewhere like alpha and tg, some not). I guess sigmas are square roots of variances and s stand for means, right? What is loc? Usually, lognormal distributions have

two parameters – mean and variance, you have three. Also, variance parameter is usually denoted as sigma squared. Substantiation of using lognormal distribution is missing.

We agree that the use of the lognormal could benefit from more justification. We chose the lognormal distribution because of its simple form and wide use describing skewed phenomena with various failure modes, including sensitivity to fungicides (Limpert, Eckhard *et al. BioScience* 51.5 (2001): 341-352.). We have now added this reference to the manuscript.

The more general three-parameter form (as discussed here: <https://www.itl.nist.gov/div898/handbook/apr/section1/apr164.htm>) includes a ‘waiting time’ or ‘shift parameter’, which is needed to describe the fact there is no germination before an onset time, and to shift the (reversed) growth rate distribution up to positive growth rate values. We have added these points and the full lognormal form (Section 2.3, Section 4.3 and Section S5).

Since our prior distributions were defined as uniform over sigma, rather than sigma squared, we think it might be more consistent to denote the parameter as sigma, though many thanks for pointing this out for future work.

4) Section 4.3 and Suppl 6: It would be nice to include some examples of the simulation images.

Thank you for suggesting this. Simulated images are plotted in Fig. 4, Fig. S5 and Fig. S6 alongside data images. However, these are quite small, and so we have now added a larger image for both to Fig. S5d for easy comparison.

5) Figure 4: Angle θ_{global} is missing in the legend. Vice versa, L is missing in the drawing – it would be nice to add – I guess it is just length from the left side of the image along medial axis of the tip, right?

Thank you for spotting this. We have added a description of θ_{global} to the Fig. 4 legend, and also drawn on L, which is the distance following the curve of the germ tube, from spore to tip.

6) Eq. 9: You defined two cases: $t > t_g$ and $t < t_g$. The case $t = t_g$ is missing.

Thank you for spotting this. We have amended such that the regime before germination includes the $t = t_g$ case (Section 2.3).

Response to Reviewer #2 (Expertise: Image-based modelling of cellular behavior, Machine/deep learning):

Cavanagh *et al.* presented a physics-informed deep-learning method to characterise the in-vitro morphological changes of the Asian soybean rust crop pathogen, *P. pachyrhizi*. Specifically, the authors applied an autoencoder model to extract salient morphological features of the single-fungus images (under different perturbation conditions), and thus created a 2D “morphospace” embedding for visualisation. They then used a Fokker Planck model to interpret the morphodynamic driving forces in the form of Waddington landscapes. Also, in combination of a random-walk-based model of the tip-growth that is fitted to the image dataset, the phenotype transitions during the growth process can be inferred.

This work aligns well with the emergent field of cellular morphological phenotyping (especially with the help of computer vision), which is gaining popularity in a wide range of disciplines, cell biology, plant biology and microbiology etc. The methodology presented in this work is generally rigorous. The results are technically sound, and demonstrated the feasibility of this integration of unsupervised learning and biophysical modelling in characterising the Asian Soybean Rust Disease, especially offering some insights into the morphological dynamics during growth - which is I believe the main contribution of the work.

Having said that, it is not clear to me, in the current form of this manuscript, how this physics-inform neural network (PINN) phenotyping strategy is superior than the existing approaches (i.e. can other methods reveal similar observations? What are their limitations?).

Thank you for suggesting a more comprehensive comparison with other methods.

Traditional agricultural phenotyping approaches analyze population level variables like mean growth rate and metabolic fluxes (something we have now mentioned in Section 1). There is currently a move in high-throughput phenotyping to implementing automated feature extraction with single-organism resolution, though with static features, as we mention in Section 1. When it comes to dynamics, there is much work quantifying animal behavior, using the principle of stereotypy (the idea that behaviour is composed of a small set of repeated patterns), as we mention in Section 1. However, these have several limitations when considering applying them to microorganism phenotyping, as we discuss in Section 1.

The most promising pipeline for agricultural morphodynamic phenotyping is Ref. [2], where prediction of insecticide modes of action targeting *C. elegans* was done using a set of thousands of behavioral features (for example 'midbody curvature during forward crawling') to cluster and classify compounds and compare the results with known modes of action, transcending a purely stereotypical behavioral analysis. However, this approach relies on thousands of behavioral features, reducing interpretability, and there is no resolution in time. It is also not applicable to destructive snapshot data using e.g. fixation.

We therefore take a different approach, seeking to infer lower-dimensional and more intuitive morphodynamic descriptors that quantify dynamics localized in time (i.e. using a model, rather than statistical correlations). We have added a description of this work, as well as limitations and benefits of our approach (Section 1).

As for the selection of the physics-informed neural network to infer the parameters: Waddington landscapes have been used to interpret dynamics with other biological data (e.g. RNA sequencing data). However, a limitation of those papers is that they solve only for steady-state distributions, where the Fokker Planck equation can be analytically solved. We selected the PINN for this challenging inverse problem with sparse (snapshot) data for the reasons now added to Sections 1 and 2.2.

Furthermore, how does this work (especially the phenotyping pipeline) can be widely applicable to large phenotypic screens of other agricultural biocides. I believe these are the key issues that the authors might want to consider to articulate in order to ensure the work is suitable for the wide readership of Nature Communications.

Thank you for suggesting the inclusion of information on how our approach could be used for other agricultural biocides. Many of the papers we cite uncover low-dimensional cell and animal morphospaces, which is the prerequisite step for our subsequent morphodynamic modelling. We then discuss potential challenges in the Discussion section (Section 3) for both the top-down Fokker-Planck model and the bottom-up mechanistic modeling approach when this shape space (morphospace) has more than two dimensions (namely increased

computation speed and reduced interpretability). However, we conclude that both analyses are still possible, and suggest methods for improving the interpretability (e.g. comparing marginal dynamics in 2D or describing the force field in terms of networks of attractors).

Apart from that, here below are some further technical questions/comments:

1. Regarding the use of autoencoder for salient feature extraction, what is the impact (is it sensitive?) on the pipeline performance if the architecture of the autoencoder is modified? Or even another type of CNN model is used for unsupervised feature extraction?

Thank you for enquiring about the sensitivity of the autoencoder's performance to changes in its structure. We have now added a discussion to the Supplementary Information (Section S4) on the significance of the autoencoder architecture, and pitfalls in changing this, as well as possible alternative algorithms, including principal component analysis (PCA), t-distributed stochastic neighbour embedding (t-SNE), and commonly used autoencoder variants that impose a Gaussian prior on the latent space, like the variational autoencoder (VAE). We also now touch on the t-SNE and PCA alternatives in the introduction (Section 1).

2. It is mentioned that the diffusion over the landscapes has two sources: morphodynamic diffusion, and embedding diffusion. It is rather unclear how to distinguish and validate the two factors.

Thank you for suggesting a more detailed explanation of the two proposed noise sources.

We have now renamed the embedding type as 'embedding noise', to highlight that this is best thought of as an 'error' that perturbs a morphology from where it would otherwise land on the 2D morphospace. For example, rotation of fungi from misalignment introduces an extra degree of freedom, which expands the dimensionality of the data manifold in the high-dimensional pixel space. A perfect autoencoder might learn to ignore such rotations and map all rotated but otherwise identical fungi to the same point in the primary 2D manifold. However, in reality this is not the case, leading to the 'embedding noise' phenomenon.

'Morphodynamic diffusion' then describes the fundamental unpredictability of dynamics on this 2D manifold. Both types are captured by the diffusion term in the Fokker Planck equation, as they deviate dynamics from being deterministic. We have added more discussion on these points to the main text (Section 2.2) and Supplementary Information (Section S4).

3. In terms of visual interpretation, I found it's rather non-trivial to relate (map) the information between the morphodynamic landscapes (e.g. Fig. S1a) and Morphospace features extracted from the decoder (e.g. Fig. S1b). It would be helpful if the authors offer a clearer visual guide in terms of the graph representation in the figure.

We agree that the correspondence between the landscapes and morphospace could be improved. We have therefore added to the Supplementary Information (Fig. S2) images of the force field with fungus images at their morphospace positions, for each condition. Since the same colouring is used for the morphospace force fields and the landscapes, hopefully the correspondence is now clearer.

4. To enable the inference of landscapes, the authors used the snapshot embeddings to infer the driving forces through a Fokker Planck model. In the model, the PINN loss (Eq. 8) is defined:

- The authors stated that "lower frequency functions are explored first". I wonder if this approach was rigorously validated in this study (although it's common to argue with "Occam's razor").

Thank you for suggesting an expansion of this point. With the new uncertainty analysis (Section 2.2 and Fig. S1), we now show how the neural network solution evolves. First, it searches low-frequency solutions, and eventually overfits to the individual snapshots (we use the early stopping regularization strategy to prevent this overfitting). We have also added a new reference (Ref. [31], cited in Section 2.2) that describes this ‘spectral bias’ of deep neural networks, whereby they prefer low-frequency solutions. In Section 2.2 we also point to the image showing the potential exploring higher frequency solutions with time (see Fig. 3c).

- Regarding the set of hyperparameters of a, b, c and d, how sensitive is the entire pipeline performance to the change of values of these parameters?

Thank you for asking about the sensitivity of the physics-informed neural network (PINN) hyperparameters used. In Section S4, we describe how we used the same hyperparameters as found to be effective when applied to the Fokker-Planck equation in the two papers cited. We then explain the reasoning in increasing the weighting for the partial differential equation (PDE) constraint: the independence of the data snapshots. We have added a quantitative vindication of this choice, showing how increasing and decreasing this weighting by an order of magnitude decreases the fit of the model with the data (in the same paragraph, in Section S4).

Response to Reviewer #3 (Expertise: Plant biophysical modeling)

The manuscript “Physics-Informed Deep Learning Characterizes Morphodynamics of Asian Soybean Rust Disease” by Cavanaugh et al. presents a deep learning based framework for extracting morphological parameters relevant to the developmental dynamics of in vitro fungal pathogen germination in response to several fungicide treatments. The framework uses automated image analysis techniques to extract static features from images, then fits a dynamic model to describe evolution with time. The framework is applied to examine an agriculturally and economically-important fungal pathogen – Asian soybean rust disease.

The primary novelty of the work lies in the combination of largely existing software components into a new overall system, and in the application to plant pathogen development. While the software developed by the authors as a whole is new, the underlying components are not necessarily novel, nor is the overall idea of dynamical, physically-based image analysis using deep learning techniques. The authors use OpenCV for static computer vision image analysis that extracts geometric features of pathogen development. The authors develop a new component for fitting of parameters in the dynamic model (i.e., physics-informed neural network; PINN) based on the existing Tensorflow 2 framework, although the authors acknowledge that other frameworks exist for neural-network based fitting of PDE’s. The in vitro imaging datasets presented are themselves novel, a limited subset of which are included in supplementary material (the authors state the full dataset can be obtained through email contact).

Thank you for mentioning the availability of the image data. We are pleased to say that all images (raw and processed) used in the study have now been deposited at <http://cellimagelibrary.org/groups/54615>.

The manuscript largely focuses on a general demonstration of the software capabilities as applied to the imaging dataset for soybean rust. A weakness of the work is that it lacks any sort of scientific hypothesis or quantitative assessment. Analysis of results is mostly

qualitative. The manuscript demonstrates that the system can identify differences in morphodynamics resulting from different fungicide treatments, but the reader is left to assume that the resulting extracted morphodynamical parameters are in reasonable accordance with reality or have any real physical relevance.

Thank you for raising the question regarding the fidelity of the morphodynamical parameters. The Fokker-Planck model is a minimal model, and as such may miss out precise attributes of the dynamics. However, the aim of this analysis is to extract intuitive morphodynamic descriptors so that ultimately hundreds of candidate compounds can be compared to gauge mode of action similarity and efficacy. Since different compounds can alter the morphodynamics, a suitable form for a more detailed model is not known *a priori*. Such detailed models would therefore be developed subsequently, after identifying promising compounds using the proposed morphodynamic screening stage. We believe the work in its current form therefore sits between 'statistical fingerprints' and physical/fully mechanistic modelling. We have expanded on the comparison with previous work in Section 1, specifically comparing with Ref. [2]. We do mention an exciting area for future work would be joint learning of the underlying morphospace and equations of motion as being currently explored for classical physics problems (see Discussion in Section 3).

It is difficult to judge the potential impact of the work. The authors make the case that analogous phenotyping technologies have resulted in significant advances in fields such as drug discovery, and similar gains could be achieved in plant pathology. While this work may be a step in the direction of achieving such a goal, I would not necessarily consider this work a substantial advance or breakthrough. Neural networks and deep learning are near the peak of the hype curve, and new applications of these approaches are being developed at an extraordinary rate in nearly every field imaginable. As I will discuss further below, I'm not convinced that such a complex deep learning based approach is needed for this particular problem in the first place.

I have listed some specific comments for the authors below:

1. Consistency of the physically-based models: It is relatively well-known that when a Langevin-based model (e.g., Eq. 1) is applied within a heterogeneous field, additional terms are needed in the Langevin equation (and in the corresponding Fokker Planck equation) in order to achieve consistent behavior. If the Langevin equation is applied in a heterogeneous field (and thus the equation coefficients are spatially variable), anomalous behavior is likely to result in which the second law of thermodynamics is violated. Without these so-called drift correction terms, entropy can decrease even in the absence of external forces. While this may appear on the surface to be a trivial issue, I would argue that there is no more important property of a dispersion model than consistency with the second law of thermodynamics, as predicted patterns resulting from a model that violates the second law may be anomalous.

This consistency condition is easy to test and I would employ the authors to do so. If a uniform distribution of particles is inserted into a field with no sources, sinks, or external forcing (i.e., $\nabla U=0$), one should find that the spatial distribution of particles remains uniform for an arbitrarily long time period. Similarly, if an initially uniform probability distribution $p(x, t_0)$ is substituted into the derived Fokker Planck equation, the resulting time rate of change of the distribution should be zero for all time ($\partial p(x,t)/\partial t = 0 \forall t$). The authors are referred to the seminal work of Thomson (1987; J. Fluid Mech. 180:529-556), which refers to this as the "well-mixed condition".

As a simple example problem, one could consider the case in which $D(x) = |0.1 \sin(2\pi x)|$ ($x \in (0,1)$) and $\nabla U=0$, an initial particle distribution that is uniformly distributed ($p(x,t_0) = \text{unif}(0,1)$), and periodic boundary conditions at $x=0$ and $x=1$. Applying Eq.

1, we will find that the particles will rapidly un-mix themselves, such that particles tend to accumulate in areas of low D , which is around $x = 0, 0.5$, and 1 . The resulting distribution of particle positions is thus anomalous, because diffusive forces can only increase entropy in which case we should get $p(x,t) = \text{unif}(0,1) \forall t$.

Thank you very much for pointing us in the direction of the interesting work in fluid dynamics on creating stochastic models in the Lagrangian frame of reference, and also for the highly intuitive example illustrating the motivation for drift-correction terms. We have read through some of the relevant parts of the fluid dynamics literature, including Thomson (1987; *J. Fluid Mech.* 180:529-556) and understand the situation as follows (please let us know if we misinterpreted the question):

Several desirable properties of the stochastic Lagrangian frame (total derivative, moving with a fluid element; e.g. Langevin equation) models of fluids had been proposed in order that they match the Eulerian frame (partial derivatives, describing bulk fluid observables; e.g. Fokker-Planck equation) observations. Thomson (1987; *J. Fluid Mech.* 180:529-556) showed that these could all be derived from a single condition: the 'well-mixed' condition. This states that particles initially well-mixed in the fluid (i.e. obeying the Eulerian distributions) at a certain point must remain so. For example, for a Lagrangian frame model with state space of position and velocity (e.g. a Langevin model), and steady-state Eulerian distributions of uniform density and Gaussian velocity, an appropriate Lagrangian frame model with initial particle distributions matching the Eulerian distributions must have the particles also following the Eulerian distributions at all following times.

In one dimension, this constraint uniquely specifies the required additional 'drift-correction' terms, but in higher dimensions there are options. This constraint therefore ultimately ensures the Lagrangian frame model is consistent with observed fluid-flow properties, including for example that it matches a uniform steady-state density profile if that is what the measurements reveal (e.g. for an incompressible fluid).

Unlike many fluid models including the Langevin model, our model has a state space of position and not both position and velocity (justification of this form is given below). We believe our model does not require extra drift-correction terms for the following reasons:

- The main purpose of the drift-correction terms is so that the Lagrangian dynamics are consistent with the Eulerian dynamics, for a set of initially consistent (i.e. well-mixed) particles. However, unlike with physical fluid models that are required to match specific flows with measured properties, there are no Eulerian distributions that the *P. pachyrhizi* morphodynamics are required to match.
- Our system describes the dynamics of living open systems for which entropy does not necessarily increase, and so an observation of decreasing entropy would not violate the second law of thermodynamics, which applies to isolated systems.
- An example of a similar minimal morphodynamic model with inhomogeneous diffusion and without drift-correction terms is a Langevin description of shape space motion of the nematode *C. elegans* (Stephens, Greg J., et al. *PNAS* 108.18 (2011): 7286-7289).

We have added a calculation of the entropy of the single-particle simulations of Eq. 1, and Supplementary Fig. 4d now shows the result that entropy always increases.

I would recommend to either re-formulate the Langevin equation to account for inhomogeneous coefficients, or simply assume that the diffusivity is spatially constant. To me, this may make more sense physically anyway, as this amounts to saying that the

pathogen development has some general time varying diffusive tendency that is augmented by a deterministic and spatially varying “growth habit” force of $-\nabla U$.

Another simple and important consistency check is to verify that an ensemble of Lagrangian particle trajectories generated using Eq. 1 produces an Eulerian probability distribution that satisfies Eq. 2. If the spatial and temporal discretization is unacceptably large, or there was an error in implementation, this can create inconsistency between Eq. 1 and Eq. 2.

Thank you for highlighting the importance of consistency between the Lagrangian and Eulerian models. If we interpret the question correctly, we already have a figure showing the agreement in Supplementary Information (Fig. S3).

2. Parsimony: Workers have been fitting Langevin and Fokker Planck models to dynamical dispersion phenomena for many decades, long before the recent popularization of neural networks and deep learning. Is such a convoluted approach needed to determine unknowns in the model, or was the PINN used because it appears more cutting-edge? Could a much simpler and easier to understand traditional statistical fitting approach be used to achieve similar results?

We absolutely agree that parsimony should be prioritised, particularly when there may be many tools for doing the same task. However, we believe that the physics-informed neural network (PINN) is a justified choice for this task at hand for the following reasons:

- When the probability density function (PDF) is known, alternative methods can be used to solve the inverse problem, however when data is only observed at discrete timepoints (as is common with biological measurement techniques that interfere with the processes under observation), PINNs are required to simultaneously infer the PDF at all times, and infer the unknown governing equation, as discussed in (Chen, Xiaoli, et al. "Solving inverse stochastic problems from discrete particle observations using the Fokker-Planck equation and physics-informed neural networks." *arXiv preprint arXiv:2008.10653* (2020).).
- PINNs can solve the inverse problem just as easily as the forward problem (all that is needed is to change which parameters are trainable), which is advantageous for interpretability and means they can solve a large class of problems.
- Grid-based solvers can typically only perform well with low-dimensional problems. However, PINNs scale far better as dimensionality increases. We aim to extend this work to higher-dimensional morphospaces in future, and as such will require this favourable scaling property.
- PINNs learn continuous fully-differentiable solutions, the benefit of which is well-described in the PINN implementation solving inverse fluid problems published in Science (Raissi, Maziar *et al.* *Science* 367.6481 (2020): 1026-1030.): “The algorithm we developed is agnostic to the geometry, initial, and boundary conditions, hence providing flexibility in choosing the domain of interest for data acquisition as well as subsequent training and predictions. Moreover, the current methodology allows us to construct computationally efficient and fully differentiable surrogates [...] that can be further used to estimate other quantities of interest [...]” For our problem, this feature is useful in transitioning between the potential and associated force field.
- Sufficiently deep neural networks can learn any function (Hornik, Kurt *et al.* " *Neural networks* 2.5 (1989): 359-366.) and are therefore highly suited to the task of function discovery. As a result of the training process exploring progressively more complex functions, we show that, using early stopping as a regularization technique, an appropriate complexity can be established from the data, without needing to guess *a priori* an appropriate complexity.

- Increasingly researchers from across different fields are becoming machine-learning literate with numerous tutorials and intuitive tools and packages. PINNs have exploded in popularity, even though they were introduced only a couple of years ago, and it's likely such tools will see increasing use in the next few decades. It therefore may be the case that deep learning tools are increasingly becoming the most widely-understood class of algorithms for such problems across disciplines.

We have added some of the points highlighted above as motivation for using a PINN to Section 2.2 (and briefly in Section 1).

3. Uniqueness: Can it be shown that the potential fields $U(x)$ and diffusivities $D(x,t)$ inferred from the PINN are mathematically unique? I am not convinced that they are, and if not it makes it difficult to infer anything physically meaningful about the resulting 'morphodynamics'.

Thank you for raising the question of uniqueness, which we agree is a challenging yet important property to investigate. This type of uncertainty is often omitted from such analyses.

We test the uniqueness of the inferred force fields (as the class of acceptable potentials is found through arbitrary shifts) and diffusivities by running the training process three times, with random mini-batches (Fig. S1, and discussed in Section 2.2 and Section S4). We then compare the spread of solutions at each spatial point, before overfitting begins (overfitting is detailed below, in relation to the question raised on the solution's convergence). This point is reached after approximately the same training time for each run. Similar values for the total loss indicate equally good solutions. This enables us to get a measure of the uncertainty in the inferred force fields and diffusivities at each point.

While it is not possible to totally rule out further solutions, we are reassured by the good coverage of data across multiple snapshots at early times and for early morphologies that there are no wildly different solutions.

4. Readers could benefit from some additional discussion regarding the reasoning behind and interpretation of the forcing function ∇U . This is essentially some non-diffusive and deterministic force that drives the spatiotemporal development of the pathogen. For a pathogen developing in vivo, this "force" would presumably also be influenced by the underlying morphology of the host, such as venation in a leaf, etc., along with a growth tendency phenotype that may or may not be manifested in vitro.

Thank you for raising these questions.

To address the concern about the physical relevance of the morphodynamical parameters, we believe we achieved the following goals: the landscapes are interpretable distillations of the dynamics hidden within numerous image sets, providing clear representations of heterogeneous phenotype transitions during growth, and differences in these across compounds. The potentials represent the deterministic part of the motion, for example extension of the germ tube from turgor pressure and vesicle delivery (the potentials largely drive in the direction of increasing length), and the diffusion then captures features that vary at single-fungus level (from both internal and external factors), including the germination time, precise growth rate, and the direction of bending. Joint modelling with internal organelles may further aid the connection of the inferred parameters with internal cellular

factors, as suggested in Discussion (Section 3). Possible connections between phenotypes and underlying molecular processes is now explained in Section S2.

What is the reasoning and justification of the form of the forcing term? This choice also means that the particle position increment dx is independent of the particle position. Why not use a form that adds correlation to particle position, such as $dx = -\nabla U \times dt + \sigma dW$?

We chose the model form (Eq. 1) as it a) is a minimal form, which aids interpretability and capturing of a wide range of dynamics and b) is the same form commonly used to quantify driving forces in biological systems, including when quantifying steady-state Waddington-type landscapes (e.g. Zhang, Feng, et al. *The Journal of chemical physics* 137.6 (2012): 065102). The driving forces in our model are position-dependent, and we do not understand the motivation for multiplying the force by the position in the morphospace.

5. Lines 35-37: Is it true that “any deep learning algorithm” relies on non-linear transformations (line 37) and uses stochastic gradient descent (line 37)? Although this may be the approach used by the authors, this is not a general description of deep learning algorithms as suggested by the preceding text.

Thank you for pointing out that these statements are very general, and that some deep learning algorithms may not rely on non-linear transformations or variants of stochastic gradient descent. We have changed this so that it now refers specifically to feedforward deep neural networks (Section 1).

6. Figure 3d: I’m struggling to formulate some type of physical interpretation of the ‘morphodynamic landscapes’ presented in Fig 3d. What does it even mean to have an asymmetric landscape? Does it make sense that the spore development, when subjected to a particular fungicide, should always “turn left”? What does it mean to have a landscape that does not achieve a zero slope at the boundary, but instead accelerates? For example with Compound X, if the trajectory makes a left-hand turn it will eventually come to equilibrium, but if it makes a right-hand turn it will accelerate and run away. I’m very skeptical of the inferred surfaces of ∇U and D , as they don’t seem to pass a sniff test. I could maybe buy the surface for DMSO given in Fig. 3d, but not so much the others.

Thank you for bringing our attention to the potentially confusing features of the inferred parameters.

The fungicides can perturb the underlying driving forces, which changes the passage of fungi moving over the morphospace. A left-turn dictated by the potential means fungi are strongly perturbed towards the leftward morphologies, depending on the diffusivity, which captures variability in the dynamics. For Compound A, for example, fungi are pushed towards the region where morphologies have multiple bends, a feature not observed in the DMSO data set (represented as a barrier for the DMSO potential).

For this phenotyping stage, we used concentrations of the compounds that generally enabled some growth (i.e. concentrations are not too high), so the features could be differentiated for better comparison of mode of action. This means that fungi are not necessarily killed. Therefore, a non-zero slope at the boundary means the fungi continue to grow past the final snapshot time, and is therefore not physically inconsistent.

The parameters outside of regions within the data domain are not constrained by the data, and we therefore included boundaries to show the regions where the probability density functions of DMSO and the fungicides falls below 10^{-3} . The regions outside of these dense

data domains are included as they aid comparison with the morphospace to localise which regions of the potentials correspond to which morphologies. We have now made these regions with low data density translucent.

I wonder if these so-called deterministic late dynamics are not statistically converged and thus determined by a small number of outlier particles? Please demonstrate statistical convergence, and if such convergence has been reached, please offer some type of physically-based explanation of this seemingly non-intuitive behavior.

Thank you for raising the important question of convergence.

We have now included a more rigorous section monitoring the convergence (Fig. S1) and showing how the PINN first learns trends common to all snapshots (due to its preference for smooth low-frequency solutions at early times, see new Ref. [31], Fig. 3c and Fig. S1c). However, ultimately it fits the individual snapshots, which are of different spore batches (since the imaging process fixates the fungi), and therefore have spurious patterns including a non-decreasing percentage of spores. Our solution is to monitor the spore region to identify when the PINN begins to fit to individual snapshots, beyond the average dynamics, and use the early stopping regularization strategy to stop training. Ultimately, the solution is to either collect more snapshots, or constrain the model physically (as we do with the tip growth model).

We feel this new section adds to the strength of the manuscript, as we (to our knowledge) are the first to describe the challenge of preventing overfitting to individual snapshots, and to propose a solution. This hopefully improves the understanding of how deep networks solve the inverse problem for the community.

We also note to the readers (in Section 2.2) that the total loss exponentially decays after an initial period of faster decay, and so reasonable results can also be achieved by stopping earlier in the training process.

Finally, the new uncertainty analysis mentioned above in relation to the solution's uniqueness shows the regions where data density is not high enough to strongly constrain the parameters (Supplementary Fig. 1).

7. Equation 1: What was the numerical timestep in integration of Eq. 1?

Thank you for suggesting the addition of the numerical timestep to the manuscript. We have now added that we used a value of 0.01 minutes (Sections 4.2 and S4).

Response to Reviewer #4 (Expertise: plant biology, mathematical modeling):

This paper shows, for the first time, a characterization of phenotypic screens, i.e. morphodynamics, of Asian soybean rust germinating in presence of different fungicides based on image sets using unsupervised methods within interpretable physical models. The method replaces human categorisation of complex phenotypes. It allows to characterizing morphodynamics (diffusion versus forced) and infers fungicide-dependent landscapes using Fokker-Planck-PDE and highlights forward simulations over the landscapes. The combined approach is highly novel and very convincing (advanced methods of deep learning, applied to important disease x fungicide-complex). I think that the knowledge gained by the proposed method will inspire further research in the highly challenging research area of deep learning and will allow further applications (e.g., in smart soil image analysis or

probably systems with more than two players).
Yet, I do have two (probably minor) concerns:

The first relates to the choice of the fungicides, i.e. to “compound X”. I am just wondering if this “syngenta research compound” would be a) freely available for future research, and, in particular b) allow supporting the reproducibility of the study - is there really no conflict of interests?

Thank you for raising concerns about including Compound X, a Syngenta research compound, in our study. We have now added the chemical structure (Fig. S7). This compound is a research intermediate and will not be marketed by Syngenta. We therefore do not believe there is a conflict of interest.

The second point relates to the model choice and further elaborations on the tip-bending model (l. 163). As far as I understood, compound A data was used for model selection (Sup Fig 3b + lines 184/185). However, the further results refer particularly to compound A data, e.g. the authors claim that “benefit of model 3 (for relaxation) is most clear for compound A” (l. 196). Well, I think this is not a good point, since model 3 was selected just because of being best performer for compound A.

Thank you for raising the problem with the wording used with this point, which we agree can be improved. We ran model selection using Compound A because this condition covered all morphologies captured under the other modelled conditions, and so model selection could be carried out on the full spectrum of morphologies (omitting Compound B and Compound C at 10mgL⁻¹ as mentioned). We just hoped to point out that the reason the model selection process then chose Model 3 was that it was the only one that could reproduce the subset of morphologies within this full spectrum that had multiple bends. We have hopefully made this more clear now (Section 2.3).

Minor remarks:

- Last paragraph in introduction: I expected to find the goals of the study (rather than the achievements, e.g. 52, 54).

Thank you for raising this point about including what the goals of the study are in the introduction, which we agree would be nice to have. We have reworded the first sentence of the final paragraph of the introduction (Section 1) to make it a statement of the goals and have also added a sentence to this paragraph mentioning the goal of interpreting the neural network convergence. We have left in the information summarising how we achieved these goals as well as the brief summary of the key results to complement this. We hope this gives the last paragraph a nice completeness and sets the scene for the rest of the manuscript.

- Fig 1a: Visualisation on top of the image snapshots in 1a: Maybe indicate what to see here?

Thank you for this suggestion. We have added a phrase directing the reader to this subfigure (Section 1), and also explained what it shows in the figure caption (of Fig. 1).

- l. 72 /73: I am not clear about the difference between compound- and condition-dependent. What are “conditions”? See also figure caption for Fig 3 “for DMSO, and the white outline shows the same but for the other conditions”.

Thank you, we agree that the difference between conditions and compounds was not clear enough in the initial manuscript. We used conditions as the more general term, capturing

changes in compounds and their concentration (because Compound C has two concentrations). We have now clarified this in Section 2.1.

- Fig 2a: I think it might be easier to grasp the idea of human versus autoencoder if this part of the figure would have been split in to a/b. What is the difference between “inset” in 2a (l 73) and “indent” in 2a (figure caption)? Probably indicate the adding value of small graph with the “lower dimensional manifolds”.

We agree that the figure could benefit from being split up more, and have partitioned the top subfigures in Fig. 2 into parts (a) and (b). We have also changed the “indent” description to “inset”.

- Fig 2: Is there a reason for indicating the spores in 2a and 2b differently (in 2b with arrow)? The figure caption should probably include details on the dashed arrows for capturing lengthening of spores and variation in shape.

We had initially not added an arrow in the first visualisation because the spores are not totally localised (images are shown on a grid, with no image exactly corresponding to the spores). However, we now added an arrow for consistency.

- Fig 3d: Are the particles on the landscapes at $t = 90$ min coloured in pink (not white)? But, I may be wrong and the tiny sphere are not representing the particles?

Yes, apologies for the error naming the colours. We have now changed it so that the legend colour corresponds correctly to the $t = 90$ min particle colour (pink).

- Eq1 vs Eq2: The last term in each equation is commutative ($Dp = pD$). Thus, why change the order?

Thank you for spotting this – we have now changed it so that the two equations have terms in the same order.

- Eq4 – Eq6: Assuming that each of these three parts of the total loss function are calculated over random minibatches (= sample), why divide by N and not by $N-1$?

If we understand the question, division by $N-1$ corrects the bias in estimating variance using a sample mean rather than true population mean. However, mean squared error losses are in machine learning divide by N as this is not a variance approximation (the target values are known).

- L. 132: Probably, should add a reference for “precluding modelling on representations found with unsupervised learning”?

Thank you for bringing our attention to this claim. Reading it back through, we feel the discussion in the previous sentence describes the meaning we intended to get across, without the addition of this general and strong claim. We have therefore removed this line from the manuscript.

- L. 134: Would it be helpful to learn more about the “threshold energy required for germination” – was not mentioned before.

We have added a comparison with an interesting paper that describes germination from a spore in fission yeast (Section 2.2).

- L 133/136: Suppl. 1a shows data for compound C, not explicitly for DMSO (yet it is mentioned there that both have large similarities).

Thank you for bringing our attention to the fact that the landscapes are not all shown in one place for clear comparison. We have now added all landscapes to Supplementary Fig. 2, with a better connection to the morphospace. Since we now use a more thorough criterion for stopping the physics-informed neural network (PINN) training (as detailed in Section 2.2 and Supplementary Fig. 1), we now see that the DMSO and Compound C at 0.041 mgL^{-1} landscapes are better differentiated, which also agrees with the results of the mechanistic tip growth model.

- L. 149: explain abbreviation “KDE”, e.g. to fig caption of Fig. 3

We have extended the abbreviation “KDE” to “kernel density estimate” for clarity, both in the Fig. 3 caption, and in the line mentioned in Section 2.2.

- L. 181: Fi4b: “for a subset of DMSO snapshots”. Why a subset? Dimension?

Thank you for pointing out that this could be made clearer. We meant that we show only 3 of the 9 snapshots to save space in the manuscript, with the dots in between these indicating there are more snapshots not shown. We have changed this phrase to hopefully make this more clear (Section 2.3 “where three of the nine DMSO snapshots are shown”, and Fig. 4 caption).

- Fig 4b: unclear about the colour code: what is grey/black and data of which compound are shown here?

- Fig 4c: The main text (l. 183) indicates it is DMSO data and not data of other compounds

Thank you for bringing to our attention the lack of clarity in the figure. Grey indicates the data images for DMSO, and black indicates the simulated images. These plots show the data-simulation comparison for DMSO and we have added a ‘DMSO’ title to the two subplots to hopefully make this clearer. We have also added a mentioned of the colour scheme to the Fig. 4 caption.

Reviewers' Comments:

Reviewer #1:

Remarks to the Author:

I am satisfied with the responses to my comments, the authors have revised the manuscript thoroughly and improved the clarity of the text as well as the transparency and reproducibility of published results. I have no further suggestions, the paper can be accepted in the current form in my opinion.

Reviewer #2:

Remarks to the Author:

I have read through the rebuttal provided by the authors and believe that they have generally addressed the key concern I raised in my previous comments. The manuscript has been adequately revised and the technical details have also been more clearly explained. Only one key concern remains: I still think that the visual presentation of the morphodynamic landscapes (Fig. 3c, d; Supplementary Fig. S2) can be further improved. For those who might not be familiar with high-dimensional phenotyping, I believe they could have a hard time understanding how to interpret the "landscape". My quick suggestions include:

- (1) a schematic drawing explaining how to interpret the landscape.
- (2) Would a 2D contour map representation be easier to interpret?
- (3) Maybe more intuitively understandable if we could change the color scale from diverging type (i.e. red-to-blue) to progressive/sequential type (from dim to bright color)?

Reviewer #3:

Remarks to the Author:

The authors of the manuscript "Physics-Informed Deep Learning Characterizes Morphodynamics of Asian Soybean Rust Disease" have, in my opinion, thoroughly and thoughtfully responded to the comments of all reviewers. There are a few points on which we can agree to disagree, but overall I feel that the manuscript is suitable for publication.

Reviewer #4:

Remarks to the Author:

This paper shows, for the first time, a characterization of phenotypic screens, i.e. morphodynamics, of Asian soybean rust germinating in presence of different fungicides based on image sets using unsupervised methods within interpretable physical models. The method replaces human categorisation of complex phenotypes. It allows to characterizing morphodynamics (diffusion versus forced) and infers fungicide-dependent landscapes using Fokker-Planck-PDE and highlights forward simulations over the landscapes. The combined approach is highly novel and very convincing (advanced methods of deep learning, applied to important disease x fungicide-complex). I think that the knowledge gained by the proposed method will inspire further research in the highly challenging research area of deep learning and will allow further applications.

The revised version looks fine to me. All my previous remarks were handled perfectly. Thank you.

Response to Reviewer #2:

I have read through the rebuttal provided by the authors and believe that they have generally addressed the key concern I raised in my previous comments. The manuscript has been adequately revised and the technical details have also been more clearly explained. Only one key concern remains: I still think that the visual presentation of the morphodynamic landscapes (Fig. 3c, d; Supplementary Fig. S2) can be further improved. For those who might not be familiar with high-dimensional phenotyping, I believe they could have a hard time understanding how to interpret the "landscape". My quick suggestions include:

- (1) a schematic drawing explaining how to interpret the landscape.
- (2) Would a 2D contour map representation be easier to interpret?
- (3) Maybe more intuitively understandable if we could change the color scale from diverging type (i.e. red-to-blue) to progressive/sequential type (from dim to bright color)?

Thank you for bringing to our attention that the visual presentation of the landscapes can be further improved for easier interpretation, and especially for the ideas on how to do this. Currently, Supplementary Fig. S2 shows the link between morphologies and landscapes. To address each of the suggestions 1, 2 and 3:

1. We have now added a schematic of Waddington's famous landscape metaphor (created by the authors in Powerpoint) to Fig. 3d, which will be familiar across disciplines and aid the interpretation of cell trajectories. We have mentioned this illustration in the caption. We have also added fungus images to some of the final cell positions, to show how ultimate fates vary, as well as 'steep' and 'shallow' labels for the colour map.
2. We have added contours to the landscapes, which we agree improve the interpretability of subtle features of the landscapes. We have kept the 3D representation as we feel this is visually stimulating and, most importantly, intuitive.
3. We implemented different colour schemes as suggested (dim to bright), however found that landscape features were harder to resolve than the 'jet' colour map currently used. Hopefully this is fine now as we have the improved interpretability from suggestions 1 & 2, while still retaining the high resolution of landscape features with the jet colourmap.